# R&D-Agent-Quant: A Multi-Agent Framework for Data-Centric Factors and Model Joint Optimization

**Yuante Li**[1]*, **Xu Yang**[2], **Xiao Yang**[2],
**Xisen Wang**[3]*, **Weiqing Liu**[2]†, **Jiang Bian**[2]
[1] Carnegie Mellon University, [2] Microsoft Research Asia, [3] University of Oxford
yuantel@cs.cmu.edu, {xuyang1, xiao.yang}@microsoft.com
xisen.wang@keble.ox.ac.uk, {weiqing.liu, jiang.bian}@microsoft.com

## Abstract

Financial markets pose fundamental challenges for asset return prediction due to their high dimensionality, non-stationarity, and persistent volatility. Despite advances in large language models and multi-agent systems, current quantitative research pipelines suffer from limited automation, weak interpretability, and fragmented coordination across key components such as factor mining and model innovation. In this paper, we propose R&D-Agent for Quantitative Finance, in short `R&D-Agent(Q)`, the first data-centric multi-agent framework designed to automate the full-stack research and development of quantitative strategies via coordinated factor-model co-optimization. `R&D-Agent(Q)` decomposes the quant process into two iterative stages: a **Research** stage that dynamically sets goal-aligned prompts, formulates hypotheses based on domain priors, and maps them to concrete tasks, and a **Development** stage that employs a code-generation agent, `Co-STEER`, to implement task-specific code, which is then executed in real-market backtests. The two stages are connected through a feedback stage that thoroughly evaluates experimental outcomes and informs subsequent iterations, with a multi-armed bandit scheduler for adaptive direction selection. Empirically, `R&D-Agent(Q)` achieves up to 2× higher annualized returns than classical factor libraries using 70% fewer factors, and outperforms state-of-the-art deep time-series models on real markets. Its joint factor–model optimization delivers a strong balance between predictive accuracy and strategy robustness. Our code is available at: https://github.com/microsoft/RD-Agent.

## 1 Introduction

Financial markets constitute high-dimensional, nonlinear dynamical systems whose return series display heavy tails [1], time-varying volatility [2], and intricate cross-sectional dependence [3]. These features imply that asset prices are driven simultaneously by macro factors, microstructural signals, and behavioral feedback [4–6], making forecasting far more challenging than conventional time series. Driven by the exponential growth of data and breakthroughs in computational power and AI techniques, the asset management industry is transitioning from experience-driven to data-driven paradigms. Within this shift, quantitative investing is becoming mainstream due to: *(i)* efficient decision-making via the data–factor–model loop, *(ii)* repeatable execution with integrated risk control, and *(iii)* precise pursuit of excess returns under increasing strategy convergence [7, 8].

Fig. 1 illustrates the modern quantitative research pipeline. Microsoft's open-source project Qlib [9] streamlines data processing and backtesting, alleviating much of the repetitive engineering burden. Consequently, this shift redirects the focus of quantitative research toward its core components: factor mining and model innovation. ***Factor mining*** progresses from

---

*Work done during an internship at Microsoft Research Asia.
†Corresponding Author

39th Conference on Neural Information Processing Systems (NeurIPS 2025) Track on Datasets and Benchmarks.

closed-form risk–return models [10, 11] such as Fama–French to evolutionary symbolic regression [12–14] and, more recently, reinforcement learning optimization of factor combinations [15–17]. *Model innovation* evolves from classical autoregression [18, 2] to machine learning models [19–21] and sequence-to-sequence deep learning architectures (e.g., GRU [22] and LSTM [23]). More recent developments include specialized time series models that decompose signals into trend–seasonal components [24] or improve attention mechanisms for long-range forecasting [25].

In parallel, stock-specific models integrate temporal event sequences with cross-sectional dependencies via graph neural networks to capture inter-stock interactions [26–28]. Recently, large language models (LLMs) and multi-agent systems further extend the information set by extracting signals from news and social networks [29–31], and simulating hedge funds and collaboration among financial experts [32–34].

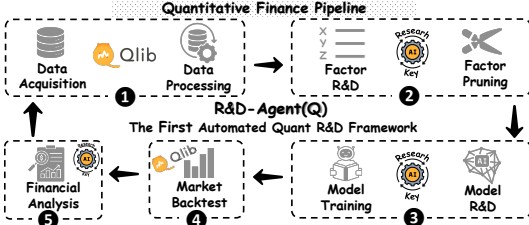

Figure 1: Quantitative finance research pipeline. Qlib makes stages ❶ and ❹ easier. `R&D-Agent(Q)` further automates stages ❷, ❸, and ❺, which are also key aspects of quantitative research.

Despite these advances, quantitative research still faces three critical limitations: *(i) Limited automation:* Current workflows require extensive human intervention in hypothesis generation, coding, and tuning, creating slow iterations and biases, Besides, semi-automated systems fail to achieve the responsiveness and scalability required for fast-moving markets. *(ii) Poor interpretability:* Existing LLM-based agents often produce trading signals directly from language interaction, without grounded factor construction or transparent model logic, and thus are prone to hallucinations. This hinders adoption in live trading, where explainability and risk controls are essential. *(iii) Fragmented optimization:* Quantitative pipelines span data processing, factor mining, model training, and evaluation, yet current approaches lack systematic task decomposition or agent-level coordination. This siloed structure limits cross-stage feedback and joint performance gains.

To address these challenges, we propose `R&D-Agent(Q)`, the first data-centric multi-agent framework for automating full-stack quantitative strategy development via coordinated factor–model co-optimization (Fig. 2). Our framework decomposes quant research into five stages spanning two core phases: **Research** and **Development**. In the Research phase, the **Specification Unit** dynamically generates goal-aligned prompts from optimization targets. The

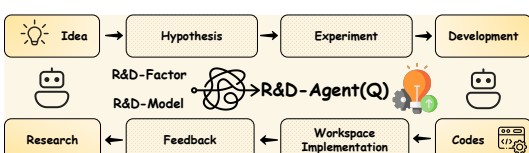

Figure 2: Conceptual diagram of `R&D-Agent(Q)`. The modules `R&D-Factor` and `R&D-Model` represent the full optimization loops for factor and model development, respectively.

**Synthesis Unit** then grows a task-specific knowledge forest from prior outcomes and generates new factor or model hypotheses, which are then mapped to executable tasks. In the Development phase, we introduce `Co-STEER`, a code-generation agent leveraging chain-of-thought [35] reasoning and a graph-based knowledge store. The **Implementation Unit** translates hypotheses into code, while the **Validation Unit** runs real-market backtests.The **Analysis Unit** evaluates with unified metrics and uses a multi-armed bandit scheduler to adaptively select the next optimization direction. This forms a closed hypothesis–implementation–validation–feedback loop that supports continual, goal-directed evolution of strategies, marking a step toward intelligent and autonomous quantitative research.

Our main contributions are as follows:

- **End-to-end automation with transparency:** `R&D-Agent(Q)` is the first data-centric multi-agent framework in quantitative finance that automates the entire R&D process with verifiable outputs that enhance interpretability and reduce hallucination risks.
- **High-performance R&D tools:** In the Research stage, `R&D-Agent(Q)` mimics analyst workflows via a structured knowledge forest, enabling the generation of coherent, high-quality hypotheses. In the Development stage, we propose `Co-STEER`, a knowledge-evolving agent tailored for data-centric tasks, improving the accuracy and efficiency of factor and model code generation.
- **Strong empirical performance:** Extensive experiments in real stock markets show that, at a cost under $10, `R&D-Agent(Q)` achieves approximately 2× higher ARR than benchmark factor libraries while using over 70% fewer factors. It also surpasses state-of-the-art deep time-series models under smaller resource budgets. Its alternating factor–model optimization further delivers excellent trade-off between predictive accuracy and strategy robustness.

## 2   R&D-Agent(Q)

Based on the formal quantitative research pipeline structure in Fig.1 and AppendixB, we propose `R&D-Agent(Q)`, a data-centric multi-agent framework for iterative factor-model R&D with automation, interpretability, and efficiency. We decompose the quantitative process into five LLM-powered units, each mainly focused on information gathering and LLM API interactions: *Specification* (scenario definition), *Synthesis* (ideas generation), *Implementation* (code development), *Validation* (backtesting), and *Analysis* (result evaluation and task scheduling). Under unified input–output constraints, these units operate in a closed-loop cycle that simulates the trial-and-error process of human quantitative researchers. Unlike manual workflows, `R&D-Agent(Q)` runs continuously and autonomously, supporting dynamic co-optimization of factor and model components. Moreover, each round's hypotheses, implementations, and results are persistently stored, enabling cumulative knowledge growth and increasingly informed decision-making over time.

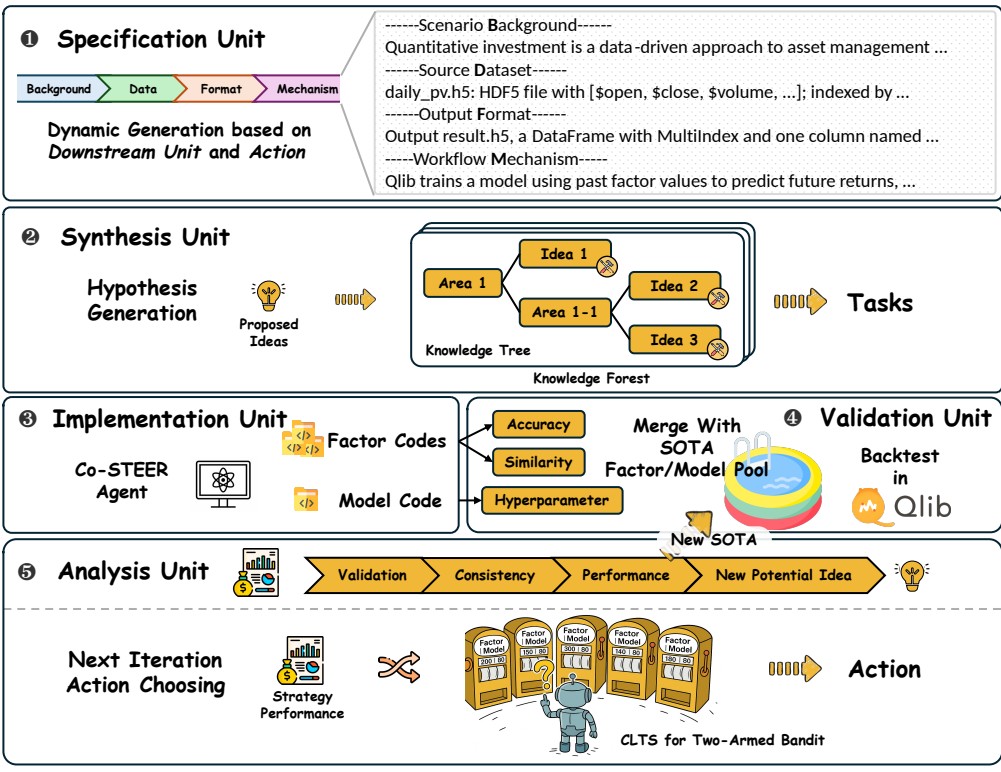

Figure 3: `R&D-Agent(Q)` consists of five functional modules that collaborate in a continuous iterative loop to generate highly effective quantitative factors and models for real-world financial markets.

### 2.1   Specification Unit

The Specification Unit serves as the top-level component of the `R&D-Agent(Q)`, dynamically configuring task context and constraints for downstream modules, ensuring consistency across design, implementation, and evaluation. It operates along two axes: ❶ Theoretical ➙ encoding prior assumptions, data schemas, and output protocols into a structured specification; ❷ Empirical ➙ establishing a verifiable execution environment and standardized interfaces for backtesting, shielding agents from low-level preprocessing and infrastructure concerns. By combining formal definitions with unified interfaces, the module reduces ambiguity and improves coordination efficiency across components.

We formalize the Specification Unit as a tuple $\mathcal{S} = (\mathcal{B}, \mathcal{D}, \mathcal{F}, \mathcal{M})$, where $\mathcal{B}$ encodes background assumptions and prior knowledge about factors or models; $\mathcal{D}$ defines the market data interface; $\mathcal{F}$ expected output format (e.g., factor tensors or return predictions); and $\mathcal{M}$ denotes the external execution environment (e.g., `Qlib`-based backtesting). Under this formulation, any candidate factor or model $f_{\boldsymbol{\theta}}$ must satisfy the condition that $\forall, x \in \mathcal{D}, ; f_{\boldsymbol{\theta}}(x) \in \mathcal{F}$ and $f_{\boldsymbol{\theta}}$ is executable within $\mathcal{M}$. This enforces compatibility with standardized input/output structures and ensures that subsequent modules can interact with $f_{\boldsymbol{\theta}}$ within a shared operational context, thereby supporting consistency and reproducibility across collaborative workflows. Implementation details are provided in Appendix E.1.

## 2.2 Synthesis Unit

The Synthesis Unit simulates human-like reasoning by generating new hypotheses based on historical experiments. Each optimization action is defined as $a_t \in \{\text{factor}, \text{model}\}$. For the current action $a_t$, the unit constructs an experiment trajectory by selecting a subset of relevant historical experiments. The $t$-th experiment is denoted by $e^t = \{h^t, f^t\}$, where $h^t$ is the hypothesis and $f^t$ is the corresponding feedback from the Analysis Unit. A set of current best-performing solutions is maintained as SOTA. Based on this, the historical hypothesis and feedback sets are defined as $\mathcal{H}_t = \{h_1, \ldots, h_t\}$ and $\mathcal{F}_t = \{f_1, \ldots, f_t\}$, respectively. Action-conditioned subsets are then extracted as Eq. (1).

$$
\begin{aligned}
\mathcal{F}_t^{(a)} &= \{f_i^a \in \mathcal{F}_t \mid a = a_t \vee e_i \in \text{SOTA}(a)\} \\
\mathcal{H}_t^{(a)} &= \{h_i^a \in \mathcal{H}_t \mid a = a_t \vee h_i \in \text{SOTA}(a)\}
\end{aligned}
\tag{1}
$$

These subsets are passed to a generative stochastic mapping $G$ (serving as the core of Research, mimicking the synthesis of theoretical priors and empirical feedback to generate valid and novel hypotheses) to produce the next hypothesis: $h^{(t+1)} = G(\mathcal{H}_t^{(a)}, \mathcal{F}_t^{(a)})$. In practice, this module relies on structured templates and standardized formats to ensure that hypotheses are both executable and scientifically grounded. For example, in a factor generation task, $h^{(t+1)}$ incorporates not only the most recent feedback but also current market conditions and domain-specific economic theory, ensuring the factor's validity and observability. To promote diversity and progressive refinement, the generation mechanism adapts its strategy based on performance feedback. If $\mathcal{F}_t^{(a)}$ suggests success, the next hypothesis increases in complexity or scope; otherwise, it undergoes structural adjustments or introduces novel variables, thereby constituting an idea forest. This adaptive mechanism enables the agent to explore new directions while maintaining responsiveness to empirical results, supporting iterative and effective strategy development.

Finally, the hypothesis $h^t$ is instantiated into a concrete task $t^t$, which the downstream implementation module uses for code-level realization. Factor hypotheses $h_t^{\text{factor}}$, due to their heterogeneity and potential interactions, can be decomposed into multiple subtasks $t_i^{\text{factor}}$. In contrast, model hypotheses, given their structural coherence, are mapped to a single task $t^{\text{model}}$ responsible for executing the entire modeling and inference pipeline.

## 2.3 Implementation Unit

The Implementation Unit is responsible for translating the executable tasks generated by the Synthesis Unit into functional code. It forms the core of complex development within the R&D-Agent(Q). To support this process, we design a specialized agent, Co-STEER, tailored for factor and model development in quantitative research. As illustrated in Fig. 4, Co-STEER integrates systematic scheduling and code-generation strategies to ensure correctness, efficiency, and adaptability in implementation.

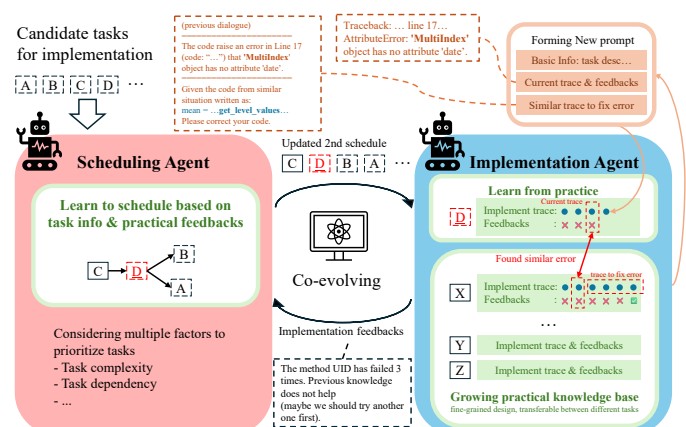

Figure 4: Diagram of the Co-STEER workflow. In the context of factor development, it consists of two main modules: the scheduling module ingests candidate tasks and performs iterative ranking based on multiple factors; the implementation module learns from previous code execution results to construct a fine-grained, transferable knowledge base applicable across tasks.

In factor development, tasks often exhibit structural dependencies. To address this, we introduce a guided chain-of-thought mechanism that encourages reasoning traceability. Specifically, the agent constructs a directed acyclic graph (DAG) $\mathcal{G} = (\mathcal{V}, \mathcal{E})$ to represent task dependencies, where an edge from task A to task B implies that A should precede B due to knowledge flow or complexity. A topological ordering $\pi_S = (t_{(1)}, \ldots, t_{(n)})$ is then derived to guide task execution. Scheduling is adaptive. Feedback from previous executions is continually integrated to improve planning: repeated failures on a task signal increased complexity, prompting prioritization of simpler tasks to enhance knowledge accumulation and execution success.

For each task $t_j$, the implementation agent $I$ generates its corresponding code $c_j$ based on both the task description and the current knowledge base, thus $c_j = I(t_j, \mathcal{K})$. This process includes task parsing, code synthesis and refinement, execution, and validation. The agent's objective is to maximize cumulative implementation quality: $\pi_I = \arg\max_\pi \mathbb{E}\left[\sum_{j=1}^{n} R_I(c_j)\right]$, where $R_I(c_j)$ evaluates the correctness and performance of code $c_j$. The knowledge base $\mathcal{K}$ plays a central role by recording successful and failed task-code-feedback triples: $\mathcal{K}^{(t+1)} = \mathcal{K}^{(t)} \cup \{(t_j, c_j, f_j)\}$, where $f_j$ denotes the feedback received after executing task $t_j$. Through a knowledge transfer mechanism, the implementation agent can also retrieve solutions to similar tasks from the knowledge base based on the current feedback $f^{(t)}$, thereby improving the efficiency and success rate of code generation for new tasks: $c_{new} = \arg\max_{c_k \in \mathcal{K}} \text{similarity}(t_{new}, t_k) \cdot c_k$. The complete algorithmic details are provided in Appendix A.1.

This feedback-driven optimization loop allows the Implementation Unit to continually enhance code quality and efficiency, facilitating rapid and robust development of quantitative research components.

## 2.4 Validation Unit

The Validation Unit evaluates the practical effectiveness of factors or model generated by the Implementation Unit. For **factors**, a de-duplication process is first applied to filter out redundant signals by computing their correlation with the existing SOTA factor library. Given the concatenated factor matrix $\mathbf{F} = [F_{\text{SOTA}}, F_{\text{new}}] \in \mathbb{R}^{T \times (M+N)}$, the IC is computed within each time slice between all $M \times N$ pairs of SOTA and new factors: $\text{IC}_{m,n}^{(t)} = \text{corr}(F_{\text{SOTA},m}^{(t)}, F_{\text{new},n}^{(t)})$, where $(m, n)$ indexes a SOTA-new pair, and $t$ indexes a time slice. These IC values are then averaged across time, and for each new factor $n$, its maximum IC across all SOTA factors is obtained as $\text{IC}_{\max}^{(n)} = \max_m \mathbb{E}_t\left[\text{IC}_{m,n}^{(t)}\right]$.

New factors with $\text{IC}_{\max}^{(n)} \geq 0.99$ are deemed redundant and excluded. After factor filtering, the remaining candidates are combined with the current SOTA model (or a baseline model, if none is available) and evaluated through the `Qlib` backtesting platform. This allows performance to be assessed under realistic market conditions. For **model**, the process is symmetric: each candidate model is paired with the current SOTA factor set and evaluated through the same backtesting pipeline.

Therefore, the Validation Unit provides an integrated and automated pipeline that supports standardized evaluation of novel components within a production-grade market simulation environment.

## 2.5 Analysis Unit

The Analysis Unit serves as both a research evaluator and strategy analyst within the `R&D-Agent(Q)` framework. After each experimental round, it conducts a multi-dimensional assessment of the current hypothesis $h^t$, the specific task $t^t$, and the experimental result $r^t$. If the experiment is judged to outperform the SOTA under action type $a_t$, its result is added to the corresponding SOTA set $\text{SOTA}(a_t)$. The unit then diagnoses failure strategy and generates targeted suggestions for refinement. The feedback $f_t$ is passed to the Synthesis Unit to guide the formulation of future hypotheses.

Notably, the Analysis Unit operates with a local view of the current experiment, while the Synthesis Unit maintains a global perspective across the full experimental history. Their interaction enables a closed-loop system that balances short-term responsiveness with long-term exploration, supporting automated iteration across research design, strategy implementation, validation, and deep analysis.

Following each analysis round, the Analysis Unit further determines whether to prioritize factor refinement or model optimization for the next iteration. To maximize performance gains, this decision is formulated as a contextual two-armed bandit problem and solved via linear Thompson sampling (see Appendix A.2 for the detailed algorithm). Specifically, At each round $t$, the system observes an 8-dimensional performance state vector $\mathbf{x}_t \in \mathbb{R}^8$, which encodes key evaluation metrics of the current strategy. The action space is $\mathcal{A} = \{\text{factor}, \text{model}\}$, corresponding to the two possible optimization paths. To evaluate the expected benefit of each action under context $\mathbf{x}_t$, we adopt a linear reward function $r = \mathbf{w}^\top \mathbf{x}_t$, where $\mathbf{w}$ reflects the relative importance of each metric. A separate Bayesian linear model is maintained for each action, with Gaussian posteriors encoding uncertainty over reward coefficients. At each step, the system samples a reward vector from each posterior and computes the corresponding expected reward. The action with the highest sampled reward is executed. After observing the actual improvement, the posterior for the chosen arm is updated. Through this contextual Thompson sampling mechanism, `R&D-Agent(Q)` adaptively balances exploration and exploitation, enabling robust performance improvement across iterations.

## 3 Experimental Setup

➥ **Datasets.** Following [36–39], we use the CSI 300 dataset, covering 300 large-cap A-share stocks in the Chinese market. The time span is split into training (Jan 1, 2008 – Dec 31, 2014), validation (Jan 1, 2015 – Dec 31, 2016), and testing (Jan 1, 2017 – Aug 1, 2020). We evaluate R&D-Agent(Q) under three configurations: ❶ R&D-Factor fixes the prediction model as LightGBM [40] and optimizes factor sets starting from Alpha 20 [3]; ❷ R&D-Model fixes the input factor set to Alpha 20 and searches for better models; ❸ R&D-Agent(Q) jointly optimizes both factor and model components.

➥ **Baselines.** At the factor level, we compare against Alpha 101 [41], Alpha 158 [42], Alpha 360 [43], and AutoAlpha [44]. At the model level, we include **machine learning models** (Linear, MLP, LightGBM[40], XGBoost [45], CatBoost [46], DoubleEnsemble [47]), and **deep learning models** (GRU [22], LSTM [23], ALSTM [48], Transformer [49], PatchTST [50], iTransformer [51], Mamba [52], TRA [38], MASTER [39], GATs [53]). More details are provided in Appendix C.3.

➥ **Evaluation Details.** We evaluate R&D-Agent(Q) using two metric categories: **factor predictive metrics**, including information coefficient (IC), IC information ratio (ICIR), rank IC, and rank ICIR; and **strategy performance metrics**, including annualized return (ARR), information ratio (IR), maximum drawdown (MDD), and Calmar ratio (CR). We follow a daily long-short trading strategy based on predicted return rankings, with position updates, holding retention rules, and realistic transaction costs. Full evaluation and implementation details are provided in Appendix C.4 and C.1.

## 4 Experiment Analysis

➥ **Analyses of Main Results.** Table 1 reports the performance of baseline models and R&D-Agent frameworks on the CSI 300 dataset, showing that the R&D-Agent consistently outperform all baselines in both predictive and strategic metrics.

❶ R&D-Factor (**Factor Optimization**). When only the factor space is adaptively optimized, both R&D-Factor$_{GPT-4o}$ and R&D-Factor$_{o3-mini}$ surpass static factor libraries (e.g., Alpha 158/360) with higher IC (up to 0.0497) and significantly improved ARR (up to 14.61%) using fewer handcrafted factors. This demonstrates that dynamic hypothesis refinement and factor screening in R&D-Agent(Q) lead to more informative signals than those from fixed, high-dimensional factor sets.

❷ R&D-Model (**Model Optimization**). For model optimization with fixed factors, R&D-Model$_{o3-mini}$ surpasses all baseline and achieves best performance on Rank IC(0.0546) and MDD ($-6.94\%$). Machine learning models lag significantly, highlighting their limitations in capturing financial noisy and non-linear patterns. While general deep learning architectures (GRU, LSTM, Transformer) deliver moderate predictive metrics, their strategic performance remains weak, suggesting a gap between feature extraction and actionable returns. Surprisingly, time-series forecasting models (e.g. PatchTST, Mamba) underperform on both fronts, indicating a fundamental mismatch between standard sequence prediction and stock market dynamics. In contrast, specialized stock prediction models (TRA, MASTER) excel in strategic metrics but trail in predictive power, highlighting a trade-off between robustness (low MDD, high IR) and precision (high IC). These results shows that adaptive model configuration—guided by automated hypothesis evaluation—yields more robust and risk-sensitive forecasting structures than both ML and handcrafted DL architectures.

❸ R&D-Agent(Q) (**Joint Optimization**). By co-optimizing factors and models, R&D-Agent(Q)$_{o3-mini}$ achieves the highest overall performance: an IC of 0.0532, ARR of 14.21%, and IR of 1.74. These improvements exceed those of the strongest baseline methods (e.g., Alpha 158, TRA) by a large margin. This demonstrates that joint refinement of factors and architectures unlocks complementary improvements, enabling scalable and consistent alpha modeling.

➥ **Analyses of Research Component.** To evaluate the research dynamics of R&D-Agent(Q), we analyze the evolution of factor hypotheses in R&D-Factor, focusing on its balance between exploration (diverse idea generation) and exploitation (local refinement). The methodology involves three steps: *(i)* **Text Embedding**: Encode generated hypothesis $h_t$ at iteration $t$ into a fixed-dimensional vector $\mathbf{h}_t$ using Sentence-BERT [54]; *(ii)* **Similarity matrix**: Compute pairwise cosine similarities to form a symmetric matrix $\mathbf{S} \in [0,1]^{T \times T}$; *(iii)* **Hierarchical Clustering**: Apply agglomerative clustering to group similar hypotheses and reorder $\mathbf{S}$ for block structure.

---

[3]twenty empirically validated factors from Alpha 158 covering momentum, value, quality, and growth.

Table 1: Experimental results of all models on the CSI 300 constituent stock dataset, including factor predictive metrics and strategy performance metrics. Visual cues indicate ranking groups: **Best** , Second Best , Good (3–8) , Average (9–14) , Poor (15–20) , and Worse (21–26) .

| Models | | CSI300 | | | | | | | |
|---|---|---|---|---|---|---|---|---|---|
| | | Factor Predictive Power Metrics | | | | Performance Metrics | | | |
| | | IC | ICIR | Rank IC | Rank ICIR | ARR | IR (SHR*) | MDD | CR |
| Machine-Learning Models | Linear | 0.0134 | 0.0992 | 0.0273 | 0.1962 | -0.0302 | -0.3710 | -0.1987 | -0.1520 |
| | MLP | 0.0291 | 0.2096 | 0.0412 | 0.3238 | 0.0003 | 0.0037 | -0.1390 | 0.0022 |
| | LightGBM | 0.0277 | 0.2211 | 0.0386 | 0.3120 | 0.0397 | 0.5664 | -0.0855 | 0.4643 |
| | XGBoost | 0.0291 | 0.2410 | 0.0384 | 0.3257 | 0.0316 | 0.4620 | -0.1139 | 0.2774 |
| | CatBoost | 0.0279 | 0.2181 | 0.0393 | 0.3110 | 0.0513 | 0.7008 | -0.0924 | 0.5552 |
| | DoubleEnsemble | 0.0294 | 0.2246 | 0.0417 | 0.3211 | 0.0551 | 0.7968 | -0.0971 | 0.5675 |
| Deep-Learning Models | Transformer | 0.0317 | 0.2538 | 0.0434 | 0.3624 | 0.0293 | 0.4267 | -0.0987 | 0.2969 |
| | GRU | 0.0315 | 0.2450 | 0.0428 | 0.3440 | 0.0344 | 0.5160 | -0.1017 | 0.3382 |
| | LSTM | 0.0318 | 0.2367 | 0.0435 | 0.3389 | 0.0381 | 0.5561 | -0.1207 | 0.3157 |
| | ALSTM | 0.0362 | 0.2789 | 0.0463 | 0.3661 | 0.0470 | 0.6992 | -0.1072 | 0.4384 |
| | GATs | 0.0349 | 0.2511 | 0.0462 | 0.3564 | 0.0497 | 0.7338 | -0.0777 | 0.6396 |
| | PatchTST | 0.0247 | 0.1945 | 0.0315 | 0.2463 | 0.0571 | 0.7191 | -0.1327 | 0.4303 |
| | iTransformer | 0.0270 | 0.1946 | 0.0340 | 0.2365 | 0.0979 | 1.2337 | -0.1151 | 0.8506 |
| | Mamba | 0.0281 | 0.2244 | 0.0374 | 0.2952 | 0.0229 | 0.3163 | -0.1154 | 0.1984 |
| | TRA | 0.0404 | 0.3197 | 0.0490 | 0.4047 | 0.0649 | 1.0091 | -0.0860 | 0.7547 |
| | MASTER | 0.0215 | 0.1925 | 0.0296 | 0.2486 | 0.0896 | 1.3406 | -0.0851 | 1.0528 |
| Factor Libraries | Alpha 101 | 0.0308 | 0.2588 | 0.0331 | 0.2749 | 0.0512 | 0.5783 | -0.1253 | 0.4085 |
| | Alpha 158 | 0.0341 | 0.2952 | 0.0450 | 0.3987 | 0.0570 | 0.8459 | -0.0771 | 0.7393 |
| | Alpha 360 | 0.0420 | 0.3290 | 0.0514 | 0.4225 | 0.0438 | 0.6731 | -0.0721 | 0.6074 |
| | AutoAlpha | 0.0334 | 0.2656 | 0.0361 | 0.2967 | 0.0400 | 0.4288 | -0.1225 | 0.3266 |
| R&D-Agent Series Framework* | R&D-Factor$_{GPT-4o}$ | 0.0489 | 0.4050 | 0.0521 | **0.4425** | **0.1461** | 1.6835 | -0.0750 | **1.9468** |
| | R&D-Factor$_{o3-mini}$ | 0.0497 | 0.3931 | 0.0500 | 0.4246 | 0.1184 | 1.3566 | -0.0910 | 1.3016 |
| | R&D-Model$_{GPT-4o}$ | 0.0326 | 0.2305 | 0.0401 | 0.2767 | 0.1229 | 1.6676 | -0.0876 | 1.4029 |
| | R&D-Model$_{o3-mini}$ | 0.0469 | 0.3688 | **0.0546** | 0.4385 | 0.1009 | 1.7009 | **-0.0694** | 1.4538 |
| | R&D-Agent(Q)$_{GPT-4o}$ | 0.0497 | 0.4069 | 0.0499 | 0.4122 | 0.1144 | 1.3167 | -0.0811 | 1.4108 |
| | R&D-Agent(Q)$_{o3-mini}$ | **0.0532** | **0.4278** | 0.0495 | 0.4091 | 0.1421 | 1.7382 | -0.0742 | 1.9150 |

Fig. 5 reveals three exploration patterns: ❶ **Local refinement followed by directional shift:** Diagonal blocks (e.g., trials 1–6, 7–11) show that R&D-Factor performs multi-step refinement within a conceptual thread before shifting direction, balancing depth with novelty. ❷ **Strategic revisitation:** Trial 26 clusters with earlier trials 12–14, demonstrating the agent's ability to revisit and incrementally refine promising early hypotheses. ❸ **Diverse paths yield synergy:** 8 out of 36 trials are selected into the final SOTA set, spanning 5 of 6 clusters. This suggests that exploring multiple directions produces complementary signals that collectively strengthen the final factor library. This *refine–shift–reuse* pattern underpins efficient deep search and broad conceptual coverage, enabling the construction of compact, diverse, and high-performing factor libraries.

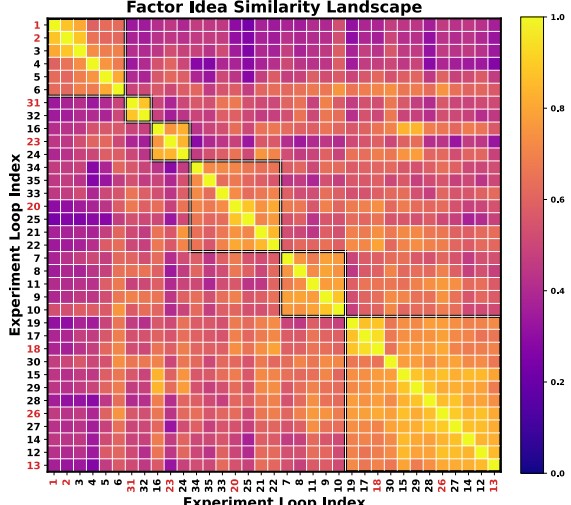

Figure 5: Cosine similarity heatmap of factor hypotheses across experiment loops in R&D-Factor. Black boxes mark clusters of similar ideas; **RED** indices indicate those selected into the SOTA factor library.

➡ **Analyses of Development Component.**
To evaluate the code generation capability of the development component, we analyze Co-STEER's performance across frameworks of R&D-Agent(Q) using the pass@$k$ accuracy metric (Fig. 6). In both factor and model tasks, success rates quickly converge within a few iterations, showing Co-STEER's ability to efficiently repair initial errors through feedback. The difference is amplified in full-stack tasks (R&D-Agent(Q)) due to their greater complexity, making iterative refinement essential. Here, o3-mini consistently achieves higher recovery rates, reflecting its stronger chain-of-thought reasoning—a clear advantage in structured, high-dependency coding scenarios. Overall,

the pass@$k$ trajectory illustrates `Co-STEER`'s ability to **progressively self-correct through iterative refinement** in structured financial coding. Additional experiments on `Co-STEER` are available in Appendix D.4.

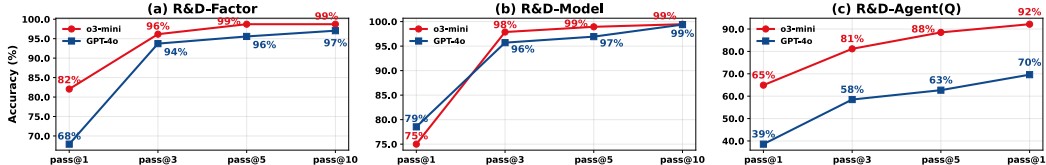

Figure 6: Pass@$k$ accuracy of `GPT-4o` and `o3-mini` in `R&D-Factor`, `R&D-Model`, and `R&D-Agent(Q)`. $x$-axis: attempts ($k$); $y$-axis: success rate within $k$ tries.

➥ **Analyses of Factor Effects.** In Fig. 7, we compare factor libraries produced by `R&D-Factor` with baselines to assess factor generation. Subfigures (a) and (b) show that even when initialized from Alpha 20, `R&D-Factor` quickly achieves IC levels comparable to Alpha 158 and Alpha 360 while using only 22% of factors. After 2017, it consistently outperforms Alpha 20, and maintains stable IC during 2019–2020 when baselines degrade. When initialized from Alpha 158, `R&D-Factor` further improves, particularly with `o3-mini`, reaching IC >0.07 in 2020 and surpassing all baselines. This demonstrates that iterative factor refinement helps **eliminate regime-sensitive or redundant signals**, improving overall predictive stability. More relevant results are provided in Appendix D.3.

➥ **Analyses of Model Effects.** Fig. 8 compares `R&D-Model` with baseline deep learning models across three dimensions: ARR, MDD, and resource usage. Both `R&D-Model` variants shift significantly toward the desirable upper-left region. `R&D-Model`$_{GPT-4o}$ achieves ARR (12%) with |MDD| (8%), attaining the highest return-risk slope. `R&D-Model`$_{o3-mini}$ offers lower drawdown with ARR (11%), yielding **strong performance under tighter risk constraints**.

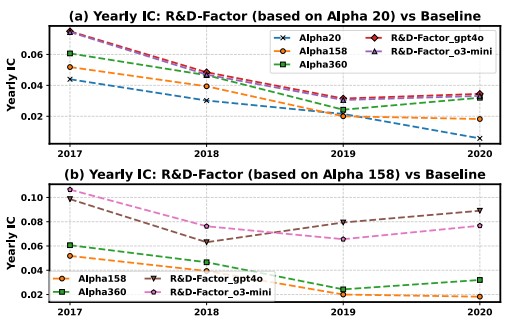

Figure 7: Comparison between classical factor libraries and `R&D-Factor`-generated factors using a LightGBM predictor on CSI 300. `R&D-Factor` was initialized from Alpha 20 or Alpha 158 and operated with `GPT-4o` or `o3-mini`.

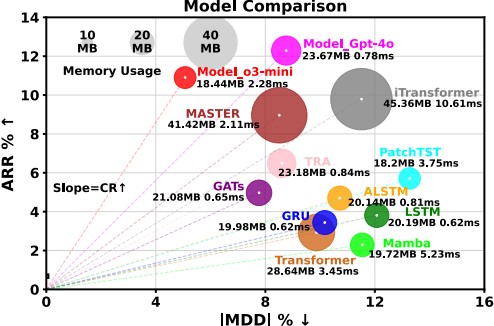

Figure 8: Comparison of models on CSI 300 in terms of return, drawdown, and resource efficiency. Bubble size encodes memory usage (MB); labels show memory and inference latency (ms). Line slope indicates Calmar Ratio.

➥ **Analyses of Empirical Scope and Generalizability.** To address concerns about scope, recency, and data leakage, we extended evaluation to another major Chinese market (CSI 500) and the U.S. market (NASDAQ 100). We adopted new dataset splits (Train: 2008-2021, Validation: 2022-2023, Test: 2024-2025), ensuring that both LLM backends have cutoffs completely or nearly prior to the test period. The detailed experimental settings and the complete results are provided in D.1.

Moreover, our framework adopts a data-centric design: the LLM is never exposed to raw market data or explicit temporal splits, but only to schema-level information (as shown in the prompt for the Specification Unit in Appendix E.1). This prevents the model from accessing precise temporal boundaries or dataset partitions, thereby mitigating the risk of information leakage.

As summarized in Table 2, `R&D-Agent(Q)` consistently achieves top-ranked performance across both Chinese and U.S. markets, with strong out-of-sample robustness in IC, ICIR, IR (SHR*), and MDD. These findings confirm that our framework generalizes beyond the Chinese market, captures recent dynamics, and remains unaffected by knowledge cutoff concerns, thereby reinforcing its **robustness** and **real-world applicability**.

Table 2: Out-of-sample experimental results on the CSI 500 and NASDAQ 100 (both tested from 2024 to June 2025), including factor predictive metrics (IC, ICIR) and strategy performance metrics (IR, MDD). Visual cues indicate ranking groups: Best, Second Best, Good (3–5), Average (6–10), Poor (11–15), and Worse (16–19).

| Models | | CSI500 | | | | NASDAQ100 | | | |
|---|---|---|---|---|---|---|---|---|---|
| | | IC | ICIR | IR (SHR*) | MDD | IC | ICIR | IR (SHR*) | MDD |
| Machine-Learning Models | LightGBM | 0.0181 | 0.1271 | -0.3178 | -0.2089 | 0.0080 | 0.0652 | -0.2603 | -0.1342 |
| | XGBoost | 0.0240 | 0.1675 | 0.0634 | -0.1766 | 0.0076 | 0.0527 | 0.1544 | -0.1211 |
| | CatBoost | 0.0241 | 0.1629 | 0.1438 | -0.1799 | 0.0095 | 0.0614 | -0.0735 | -0.1148 |
| | DoubleEnsemble | 0.0248 | 0.1705 | 0.2500 | -0.2094 | 0.0047 | 0.0360 | -0.0046 | -0.1404 |
| Deep-Learning Models | Transformer | 0.0194 | 0.1355 | 0.2898 | -0.1331 | -0.0011 | -0.0077 | -0.0343 | -0.1553 |
| | GRU | 0.0188 | 0.1022 | 0.3716 | -0.1602 | 0.0064 | 0.0457 | 0.2930 | -0.1504 |
| | LSTM | 0.0219 | 0.1434 | 0.6900 | -0.1075 | 0.0062 | 0.0409 | 0.4526 | -0.1204 |
| | GATs | 0.0162 | 0.1013 | 0.5168 | -0.1569 | -0.0004 | -0.0023 | 0.5772 | -0.1491 |
| | iTransformer | 0.0161 | 0.1031 | 0.0985 | -0.1496 | 0.0076 | 0.0421 | 0.3612 | -0.1991 |
| | TRA | 0.0260 | 0.1813 | 0.6040 | -0.1461 | 0.0058 | 0.0446 | 0.4608 | -0.1351 |
| Factor Libraries | Alpha 158 | 0.0192 | 0.1353 | 0.2515 | -0.1771 | 0.0040 | 0.0324 | 0.0303 | -0.1140 |
| | Alpha 360 | 0.0195 | 0.1331 | 0.2527 | -0.1270 | 0.0042 | 0.0327 | 0.5890 | -0.1182 |
| | AutoAlpha | 0.0184 | 0.1529 | 0.5728 | -0.1006 | 0.0046 | 0.0265 | 0.0974 | -0.1165 |
| R&D-Agent Series Framework* | R&D-Factor (GPT-4o) | 0.0201 | 0.1709 | 1.3730 | -0.0787 | 0.0070 | 0.0446 | 1.0985 | -0.0977 |
| | R&D-Factor (o4-mini) | 0.0264 | **0.2652** | 1.0014 | -0.1215 | 0.0166 | 0.1017 | 1.1169 | -0.0650 |
| | R&D-Model (GPT-4o) | 0.0259 | 0.1649 | 1.0941 | -0.1367 | 0.0128 | 0.0831 | 1.0742 | -0.0842 |
| | R&D-Model (o4-mini) | 0.0265 | 0.1825 | 1.4021 | -0.0735 | 0.0081 | 0.0484 | 1.2671 | -0.0741 |
| | R&D-Agent(Q) (GPT-4o) | 0.0241 | 0.1532 | 1.4227 | -0.0803 | **0.0172** | 0.0908 | 1.3312 | -0.1044 |
| | R&D-Agent(Q) (o4-mini) | **0.0288** | 0.1828 | **2.1721** | **-0.0656** | 0.0162 | **0.1035** | **1.7737** | **-0.0634** |

➡ **Ablation Study.** To evaluate the impact of different action selection strategies, we conduct an ablation study as shown in Table 3. The Bandit scheduler achieves the best overall performance, with the highest IC, ARR, and number of SOTA selections, confirming its ability to **prioritize the most promising optimization targets under limited computational budgets**. The LLM-based strategy performs moderately but incurs higher per-step overhead due to additional model calls, resulting in fewer iterations. Random scheduling performs the worst, underscoring the importance of informed decision-making in driving effective optimization. Full ablation results are provided in Appendix D.2.

Table 3: Ablation results on action selection strategies for `R&D-Agent(Q)` (o3-mini). We compare random, LLM-based, and Bandit controllers across predictive quality, strategy performance, and execution statistics (TL: total loops, VL: valid loops, SL: SOTA selections, TRH: runtime in hours).

| Models | | Factor Predictive Power Metrics | | Performance Metrics | | Execution Metrics | | | |
|---|---|---|---|---|---|---|---|---|---|
| | | IC | ICIR | ARR | MDD | TL | VL | SL | TRH |
| Algorithm Ablation | `R&D-Agent(Q)`w/ random | 0.0445 | 0.3589 | 0.0897 | -0.1004 | 33 | 19 | 7 | 12 |
| | `R&D-Agent(Q)`w/ LLM | 0.0476 | 0.3891 | 0.1009 | -0.0794 | 33 | 20 | 5 | 12 |
| | `R&D-Agent(Q)`w/ Bandit | **0.0532** | **0.4278** | **0.1421** | **-0.0742** | **44** | **24** | **8** | 12 |

➡ **Backend Comparisons.** To assess the sensitivity of `R&D-Agent(Q)` to the LLM backend, we evaluate six API variants across research and strategy metrics (Fig. 9). Despite moderate loop statistics, `o1` achieves top performance through several impactful rounds with strong strategic breakthroughs. The recently released `GPT-4.1` ranks second across most metrics. Other variants (except `GPT-4o-mini`, with limited reasoning capacity, leading to weaker performance) exhibit comparable results, showing the robustness of our framework across LLM backends.

➡ **Extended Studies.** Appendix D.5 further shows that `R&D-Agent(Q)`'s cost (in the paper's setting) is under $10, confirming its cost-efficient scalability. Appendix D.6 validates its robustness on real-world quant scenarios.

Figure 9: Comparison of `R&D-Agent(Q)` using different API backends (30 loops each). Axes are normalized, with outer points indicating better performance.

## 5 Related Work

**Traditional Methods in Quantitative Research.** Quantitative strategies have traditionally relied on human-crafted factors from asset pricing theory, such as value and momentum [10, 11]. While

interpretable, these fixed signals often lack flexibility in adapting to changing regimes. To overcome these limitations, symbolic regression and genetic programming (GP) methods [14, 55] automate factor mining by evolving complex, non-linear expressions. Enhancements like lag operators [13] and operator mutation with pruning [56] yield more diverse and effective signals. Reinforcement learning (RL) methods reframe factor allocation as sequential decision-making, directly optimizing Sharpe or Calmar ratios [15, 57]. Andre et al. [16] model factor weights via Dirichlet policies with KL regularization, enabling sparse and adaptive strategies. However, RL methods often lack robustness under regime shifts (e.g., 2020 circuit breaker [58]) and remain hard to interpret.

Model-wise, early approaches like ARIMA [59] and exponential smoothing [60] struggle with noisy, high-dimensional data. Classical machine learning methods (e.g., SVMs [19], random forests [20]) improve robustness but still require manual feature engineering. Deep learning models like LSTMs [61] and Transformers [62] have since been applied to capture long-term and cross-sectional dependencies [63, 64]. Building upon these, specialized time series neural networks have emerged. PatchTST [65] segments inputs into local patches, while iTransformer [66] remaps variable-token relations to model multivariate structure. Domain-specific models like MASTER [67] further incorporates market-level dynamics for improved financial prediction. However, both factor and model pipelines remain siloed, expert-dependent, and inflexible, restricting scalability in volatile markets.

**LLM-Driven Agents in Finance.** Large language models (LLMs) offer new opportunities for automating financial research due to their strong reasoning and abstraction capabilities. Recent work explores their use in extracting predictive signals from financial text [68, 31], generating factor explanations [30], and enabling multi-modal market analysis [33]. Parallel advances in LLM-based multi-agent systems (e.g., AutoGen [69], AutoGPT [70]) provide coordination frameworks for complex decision-making. In finance, systems like FinAgent [33] and TradingAgents[34] use role-based agents for subtasks such as event extraction or portfolio updates. However, most existing efforts focus on narrow subtasks and rely heavily on semantic signals, making them prone to hallucination, hard to interpret, and difficult to reproduce. Moreover, they lack mechanisms for joint factor-model optimization or workflow integration, limiting their effectiveness in real-world quantitative systems.

## 6   Conclusion

We propose `R&D-Agent(Q)`, a LLM-driven framework for collaborative factor-model development in quantitative finance. By decomposing research into modular components and integrating a bandit-based scheduler, it supports efficient, adaptive iteration under fixed compute budgets. Empirically, `R&D-Agent` outperforms baselines in both signal quality and strategy performance, with strong cost-efficiency and generalizability. Its modularity also enables adaptation to real-world settings. However, current framework relies solely on the LLM's internal financial knowledge. Future work may enhance data diversity, incorporate domain priors, and enable online adaptation to evolving market regimes.

## 7   Disclaimer

Users of the `R&D-Agent(Q)` framework and the associated code should prepare their own financial data and independently assess and test the risks of the generated factors and model in use's own scenarios. It is essential to use the agent-generated code, data, and model with caution and thoroughly check them. The `R&D-Agent(Q)` framework does not provide financial opinions, nor is it designed to replace the role of qualified financial professionals in formulating, assessing, and approving financial products. The outputs of the `R&D-Agent(Q)` framework do not reflect the opinions of Microsoft.

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

# A  Algorithmic Details

## A.1  Co-STEER Implementation Logic

To further clarify the internal mechanism of the `Co-STEER` agent, we provide both a formal algorithmic procedure (Algorithm 1) and a comparative method analysis (Table 4).

Existing natural-language-to-code methods primarily focus on isolated capabilities. Few-shot [71] learning leverages in-context examples to guide model outputs. Chain-of-Thought (CoT) [35] improves reasoning coherence via step-by-step prompting. Reflexion [72] and Self-Debugging [73] emphasize iterative correction through feedback, while Self-Planning [74] supports automatic task decomposition.

In contrast, `Co-STEER` offers a unified solution that integrates scheduling, reasoning, feedback-driven refinement, and long-term knowledge accumulation. It maintains a continually growing knowledge base of prior attempts, which enables retrieval and adaptation of previously successful solutions. Additionally, its scheduling agent supports multi-task code generation, such as factor implementation in quantitative research, by dynamically prioritizing tasks based on complexity and feedback—favoring simpler, more foundational tasks early on to provide informative scaffolding for subsequent code generation.

Table 4: Comparison of code-generation methods in quantitative research. `Co-STEER` is the only method supporting end-to-end development, from task scheduling to implementation, enhanced by structured reasoning and lifelong knowledge reuse. This makes it especially suited for multi-step, data-centric financial workflows.

| Methods | Schedule | Implementation | | | |
|---|---|---|---|---|---|
| | - | Demonstration | Planning or Reasoning Before Impl. | LLM-Based Self-Feedback | Growing Practical Knowledge |
| Few-shot [71] | ✗ | ✓ | ✗ | ✗ | ✗ |
| CoT [35] | ✗ | ✗ | ✓ | ✗ | ✗ |
| Reflexion [72] | ✗ | ✗ | ✗ | ✓ | ✗ |
| Self-Debugging [73] | ✗ | ✗ | ✗ | ✓ | ✗ |
| Self-Planning [74] | ✗ | ✗ | ✓ | ✗ | ✗ |
| Co-STEER | ✓ | ✓ | ✓ | ✓ | ✓ |

Below, we present the complete pseudocode of the Co-STEER workflow for the factor implementation task.

## A.2  Bandit Scheduling Logic

As stated in Section 2.5, we apply contextual Thompson Sampling to adaptively select between two optimization directions—factor refinement and model optimization—based on current strategy performance. The problem is formulated as a two-armed contextual bandit with linear reward functions. Each arm maintains its own Bayesian linear regression model, whose posterior is updated after every interaction.

At each round $t$, the system summarizes strategy quality using the following 8-dimensional performance vector:

$$\mathbf{x}_t = \begin{bmatrix} \text{IC, ICIR, Rank(IC), Rank(ICIR), ARR, IR,} -\text{MDD, SR} \end{bmatrix}^\top \in \mathbb{R}^8$$

Each component is positively correlated with desirable strategy outcomes; maximum drawdown (MDD) is negated to align with this direction.

Given a prior $\mu^{(a)} = \mathbf{0}$, $P^{(a)} = \tau^{-2}I$, the algorithm samples a reward coefficient vector from the posterior for each action, estimates the reward under the current context $\mathbf{x}_t$, and selects the action with the highest sampled value. The posterior is then updated using standard Bayesian linear regression updates.

**Algorithm 1** Co-STEER: Collaborative Scheduling and Task Execution Engine for Quant Research
___

**Require:** Tasks $\mathcal{T} = \{t_1, \ldots, t_n\}$, Knowledge base $\mathcal{K}$
**Ensure:** Implemented code solutions $\{c_1, \ldots, c_n\}$
1: Initialize DAG $\mathcal{G} = (\mathcal{V}, \mathcal{E})$ where $\mathcal{V} = \mathcal{T}$
2: Initialize task complexity scores $\alpha_j = 1$ for all $t_j \in \mathcal{T}$
3: **function** UPDATETASKORDER($\mathcal{G}, \{\alpha_j\}$)
4:     Compute weighted edges: $w_{ij} = \alpha_i/\alpha_j$ for $(i, j) \in \mathcal{E}$
5:     Return topological order $\pi_S$ considering $w_{ij}$
6: **end function**
7: **function** IMPLEMENTTASK($t_j, \mathcal{K}, f^{(t)}$)
8:     Find similar tasks: $S_j = \{t_k \in \mathcal{K} : \text{similarity}(t_j, t_k) > \theta\}$
9:     $c_{ref} = \arg\max_{c_k \in \mathcal{K}} \text{similarity}(t_j, t_k) \cdot c_k$
10:     Generate code: $c_j = I(t_j, c_{ref}, \mathcal{K})$
11:     Execute and get feedback $f_j$
12:     Update knowledge base: $\mathcal{K} \leftarrow \mathcal{K} \cup \{(t_j, c_j, f_j)\}$
13:     **return** $(c_j, f_j)$
14: **end function**
15: **while** $\mathcal{T}$ not empty **do**
16:     $\pi_S \leftarrow$ UPDATETASKORDER($\mathcal{G}, \{\alpha_j\}$)
17:     **for** $t_j \in \pi_S$ **do**
18:         $(c_j, f_j) \leftarrow$ IMPLEMENTTASK($t_j, \mathcal{K}, f^{(t)}$)
19:         **if** $f_j$ indicates failure **then**
20:             Update complexity: $\alpha_j \leftarrow \alpha_j + \delta$
21:             Break and recompute $\pi_S$
22:         **else**
23:             $\mathcal{T} \leftarrow \mathcal{T} \setminus \{t_j\}$
24:         **end if**
25:     **end for**
26: **end while**
27: **return** $\{c_1, \ldots, c_n\}$
___

**Algorithm 2** Contextual Thompson Sampling scheduler used to adaptively choose between factor and model optimization.
___

    *(Assume prior: $\mu^{(a)} = 0$, $P^{(a)} = \tau^{-2}I$)*
1: Define reward weight vector $\mathbf{w} \in \mathbb{R}^8$
2: **for** $t = 1$ **to** $T$ **do**
3:     Get performance state vector $\mathbf{x}_t$
4:     **for all** $a \in \mathcal{A}$ **do**
5:         Sample $\tilde{\boldsymbol{\theta}}^{(a)} \sim \mathcal{N}(\mu^{(a)}, (P^{(a)})^{-1})$
6:         Compute expected reward: $\hat{r}^{(a)} = \tilde{\boldsymbol{\theta}}^{(a)\top} \mathbf{x}_t$
7:     **end for**
8:     Select $a_t = \arg\max_a \hat{r}^{(a)}$; observe reward: $r_t$
9:     Update $P^{(a_t)}$ and $\mu^{(a_t)}$ based on $(\mathbf{x}_t, r_t)$

$$P^{(a_t)} \leftarrow P^{(a_t)} + \frac{1}{\sigma^2} \mathbf{x}_t \mathbf{x}_t^\top$$

$$\mu^{(a_t)} \leftarrow \left(P^{(a_t)}\right)^{-1} \left(P^{(a_t)} \mu^{(a_t)} + \frac{1}{\sigma^2} r_t \mathbf{x}_t\right)$$

10: **end for**
___

## B Formal Definition of the Quantitative Research Pipeline

Building on several classic works in quantitative finance [9, 39, 75], we define the raw dataset as a three-dimensional tensor with dual temporal indexing, denoted as $\mathbf{X} \in \mathbb{R}^{N \times T \times P}$. This dataset takes stocks as the underlying asset class, each associated with a set of factor dimensions. As formally defined in Equation (2), the tensor contains $N$ assets over an observation period $\mathcal{T} = \{1, \ldots, T\}$. The row index corresponds to time $t$, the column index to asset $i$, and the channel index $p$ refers to one of the $P$ factor dimensions. Each entry $x_{i,t}^{(p)}$ represents the value of the $p$-th factor for asset $i$ at time $t$.

$$\mathbf{X} = \{x_{i,t}^{(p)} \mid i \in [N], t \in \{1, \ldots, T\}, p \in [P]\} \in \mathbb{R}^{N \times T \times P}, \tag{2}$$

New factors are then constructed either analytically or using machine learning, based on the original factor features. Given a sliding window of length $\ell$, $m$ new factors $f_{i,t}^{(j)}$ are generated via a mapping defined in Equation (3), where $\mathbf{x}_{i,t-\ell+1:t}$ denotes the recent $\ell$-day tensor slice for asset $i$, and $f_{i,t}^{(j)}$ is the $j$-th derived factor. The new factor tensor $\mathbf{Z} \in \mathbb{R}^{N \times T \times (P+m)}$ is formed by concatenating the original and generated factors, with $\mathbf{z}_{i,t}$ representing the concatenated vector at $(i, t)$, as shown in Equation (4).

$$\Phi : \mathbb{R}^{\ell \times P} \longrightarrow \mathbb{R}^m, \quad \Phi(\mathbf{x}_{i,t-\ell+1:t}) = (f_{i,t}^{(1)}, \ldots, f_{i,t}^{(m)}), \tag{3}$$

$$\mathbf{z}_{i,t} = [\mathbf{x}_{i,t} \| \mathbf{F}_{i,t}] \in \mathbb{R}^{P+m}, \quad \mathbf{F}_{i,t} = \Phi(\mathbf{x}_{i,t-\ell+1:t}). \tag{4}$$

Given the noisy and incomplete nature of financial data, a two-stage preprocessing procedure is adopted to suppress the impact of outliers. First, a cross-sectional robust Z-score normalization is applied to each feature, as in Equation (5), where MAD is the median absolute deviation and $\varepsilon$ is a numerical stability constant. Second, missing values are imputed using a "forward-fill + cross-sectional mean" strategy, formally defined in Equation (6).

$$\tilde{x}_{i,t}^{(p)} = \frac{x_{i,t}^{(p)} - \text{Median}_i\left(x_{\cdot,t}^{(p)}\right)}{\text{MAD}_i\left(x_{\cdot,t}^{(p)}\right) + \varepsilon}, \tag{5}$$

$$x_{i,t}^{(p)} \leftarrow \begin{cases} x_{i,t-1}^{(p)}, & \text{if } x_{i,t}^{(p)} = \text{NA and } x_{i,t-1}^{(p)} eq \text{NA}, \\ \text{Mean}_i\left(x_{\cdot,t}^{(p)}\right), & \text{otherwise.} \end{cases} \tag{6}$$

Asset returns are defined as the prediction target for training and validation, denoted $y_{i,t}^{(\tau)}$ with $\tau = 1$ trading day in this work, as shown in Equation (7). Similar to factor preprocessing, missing labels are removed, and Z-score normalization is applied cross-sectionally on each trading day, yielding the supervised sample pairs $\left(\mathbf{z}_{i,t}, \tilde{y}_{i,t}^{(\tau)}\right)$.

$$y_{i,t}^{(\tau)} = \frac{P_{i,t+\tau} - P_{i,t}}{P_{i,t}}, \quad P_{i,t} \text{ is the closing price of asset } i \text{ at time } t, \tag{7}$$

$$\tilde{y}_{i,t}^{(\tau)} = \frac{y_{i,t}^{(\tau)} - \text{Mean}_i\left(y_{\cdot,t}^{(\tau)}\right)}{\text{Std}_i\left(y_{\cdot,t}^{(\tau)}\right) + \varepsilon}, \tag{8}$$

To better encapsulate the interaction interface between factors and models, predictions $\hat{y}_{i,t}$ are uniformly generated by a predictor $f_{\boldsymbol{\theta}}$ as defined in Equation (9), where $\boldsymbol{\theta}$ denotes learnable parameters. The predictor supports both tabular models (which take $\mathbf{z}_{i,t}$ as input) and time series models, which instead use the tensor slice $\mathbf{Z}_{i,t-\ell+1:t}$ to capture temporal structures.

$$\hat{y}_{i,t} = f_{\boldsymbol{\theta}}(\mathbf{z}_{i,t}), \quad f_{\boldsymbol{\theta}} : \mathbb{R}^{P+m} \to \mathbb{R}, \tag{9}$$

Model training follows a walk-forward validation procedure, minimizing the mean squared error loss $\mathcal{L}(\boldsymbol{\theta})$ (Equation (10)) via gradient descent to optimize parameters to $\boldsymbol{\theta}^*$.

$$\mathcal{L}(\boldsymbol{\theta}) = \frac{1}{|\mathcal{D}_{\text{train}}|} \sum_{(i,t) \in \mathcal{D}_{\text{train}}} \left( y_{i,t}^{(\tau)} - \hat{y}_{i,t} \right)^2. \tag{10}$$

This pipeline covers four essential components—data representation, feature engineering, sample construction, and model training—providing a standardized input interface for the dual-loop factor-model optimization mechanism in the `R&D-Agent(Q)` framework.

## C  Experimental Details

### C.1  Implementation Settings

**Hardware Setup.** All experiments were conducted on a dedicated server equipped with dual Intel Xeon Gold 6348 CPUs, providing a total of 112 threads, and four NVIDIA RTX A6000 GPUs, each with 48 GiB of memory (192 GiB in total).

**Evaluation Protocol.** All models were trained and validated on consistent dataset splits to ensure fair comparison. For each baseline model, we conducted extensive hyperparameter tuning and evaluated robustness using five independent runs with different random seeds. Rather than reporting standard error bars or confidence intervals, we report the **median** annualized return (ARR) across these runs. This follows quantitative finance practice, where consistent outperformance is valued more than pointwise variance. The use of median also reduces the influence of outliers.

**Framework Configuration.** The experiments were conducted using the `R&D-Agent` framework. `R&D-Factor` automates factor design and evaluation, and `R&D-Model` optimizes predictive models. Each module runs for 6 hours per experiment. The joint framework `R&D-Agent(Q)` alternates between the two for 12 hours total. Persistent caching was enabled throughout to accelerate repeated access to intermediate outputs, including the SOTA factor library, which is referenced in each loop.

**Parameter Configuration.** We used various LLM API backends during experiments. For `GPT-4o` and `GPT-4o-mini`, we enabled streaming, set temperature to 0.8, and capped token usage at 4096. For `o3-mini`, `o3`, and `GPT-4.1`, we disabled streaming, used a temperature of 1.0, and allowed up to 10,000 tokens. All API interactions used a unified system prompt from the user role, adjusted per backend capability. In the `Co-STEER` module (Implementation Unit), we used `text-embedding-ada-002` to compute semantic embeddings for code, hypothesis, and log analysis. To ensure efficiency in fine-grained debugging, inner optimization loops in `Co-STEER` were capped at 10 iterations per task for both factor and model workflows. For each task, we set the maximum runtime of the Implementation Unit to 600 seconds, and the Validation Unit to 3600 seconds.

### C.2  Dataset

We do not propose a new dataset in this paper. The baseline factor library, Alpha 20, is shown in Table 5. The raw financial data used for factor mining can be divided into two categories: market data and fundamental data. The market data can be generated using the script at `https://github.com/microsoft/RD-Agent/blob/main/rdagent/scenarios/qlib/experiment/factor_data_template/generate.py`, while the fundamental data—which includes standard financial indicators such as profitability, valuation, and growth metrics—is listed in Table 6.

Table 6: Fundamental data fields used for factor mining in `R&D-Agent(Q)`. These fields can be obtained by searching their names in the `Wind` terminal (`https://www.wind.com.cn/mobile/WFT/en.html`).

| Factor | Description |
|--------|-------------|
| **Profitability** | |
| ROE_TTM | Return on Equity (TTM) |
| ROA_TTM | Return on Assets (TTM) |
| ROIC | Return on Invested Capital |
| EBIT_EV | EBIT / Enterprise Value |

| Factor | Description |
| --- | --- |
| EBITDA_EV | EBITDA / Enterprise Value |
| NET_PROFIT_YOY | Net Income Year-over-Year Growth |
| NET_PROFIT_YOY_Q | Net Income YoY Growth (Quarterly) |
| NET_PROFIT_MARGIN | Net Profit Margin |
| NET_PROFIT_MARGIN_TTM | Net Profit Margin (TTM) |
| GROSS_PROFIT_MARGIN_TTM | Gross Profit Margin (TTM) |

## Growth

| Factor | Description |
| --- | --- |
| NET_PROFIT_YOY_TTM | Net Profit Growth (TTM) |
| OPER_REV_YOY_TTM | Revenue Growth (TTM) |
| OPER_REV_YOY | Revenue Year-over-Year Growth |
| OPER_PROFIT_YOY | Operating Profit Year-over-Year Growth |
| OPER_REV_YOY_Q | Revenue YoY Growth (Quarterly) |
| OPER_REV_QOQ | Revenue Quarter-over-Quarter Growth |
| OPER_PROFIT_QOQ | Operating Profit QoQ Growth |
| NET_PROFIT_QOQ | Net Profit QoQ Growth |
| EST_OPER_REV_CHANGE_1M | Forecast Revenue Change (1M) |
| EST_OPER_REV_CHANGE_3M | Forecast Revenue Change (3M) |

## Valuation

| Factor | Description |
| --- | --- |
| EP_TTM | Earnings-to-Price Ratio |
| BP | Book-to-Price Ratio |
| EP_FY1 | Forward Earnings-to-Price Ratio |
| BP_FY1 | Forward Book-to-Price Ratio |
| CFO_TO_PRICE_TTM | Cash Flow-to-Price Ratio |
| OCF_TO_MKT_CAP | Operating Cash Flow / Market Cap |
| OCF_TO_PRICE_TTM | Operating Cash Flow-to-Price Ratio |

## Risk & Volatility

| Factor | Description |
| --- | --- |
| VOLATILITY_1M | 1-Month Volatility |
| VOLATILITY_3M | 3-Month Volatility |
| VOLATILITY_1Y | 12-Month Volatility |
| BETA_1Y | Market Beta (12-Month) |
| IDIOSYNCRATIC_VOLATILITY | Idiosyncratic Volatility |

## Risk & Volatility

| Factor | Description |
| --- | --- |
| VOLATILITY_1M | 1-Month Volatility |
| VOLATILITY_3M | 3-Month Volatility |
| VOLATILITY_1Y | 12-Month Volatility |
| BETA_1Y | Market Beta (12-Month) |
| IDIOSYNCRATIC_VOLATILITY | Idiosyncratic Volatility |

## Quality

| Factor | Description |
| --- | --- |
| INTEREST_COVERAGE | Interest Coverage Ratio |
| CFO_TTM | Cash Flow from Operations (TTM) |
| CFO_Q | Cash Flow from Operations (Quarterly) |
| CFO_TO_OPER_REV_TTM | Operating Cash Flow / Revenue (TTM) |
| NET_PROFIT_MARGIN | Net Profit Margin |
| ASSET_TURNOVER_TTM | Asset Turnover (TTM) |
| FIXED_ASSET_TURNOVER_TTM | Fixed Asset Turnover (TTM) |

## Sentiment & Flow

| Factor | Description |
| --- | --- |
| MID_ORDER_BUY_AMT | Medium Order Active Buy Amount |
| MID_ORDER_SELL_AMT | Medium Order Active Sell Amount |
| RATING_UPGRADE | Analyst Rating Upgrade |
| RATING_DOWNGRADE | Analyst Rating Downgrade |
| ANALYST_MOMENTUM_SCORE | Analyst Momentum Score |

| Factor | Description |
|--------|-------------|
| **Momentum** | |
| RETURN_1M | 30-Day Return |
| RETURN_2M | 60-Day Return |
| RETURN_3M | 90-Day Return |
| RETURN_6M | 182-Day Return |
| RETURN_1Y | 365-Day Return |

## C.3 Baselines

In the benchmark experiments, factor-based experiments are conducted using the LightGBM model, and model-based experiments are conducted using the Alpha 20 factor library.

**Machine Learning Models.**

- **Linear Model**: The most basic linear regression model, used to model linear relationships between features and the target variable. It is highly interpretable and has low complexity, serving as the lower bound benchmark for model predictive performance.

- **Multilayer Perceptron (MLP)**: A feedforward neural network architecture that includes one or more nonlinear hidden layers, suitable for modeling nonlinear relationships between features.

- **LightGBM** [40]: A tree-based model built on the gradient boosting framework. It uses histogram-based split methods and a leaf-wise growth strategy, offering fast training speed and low memory usage. Source code is available at: `https://github.com/microsoft/LightGBM`.

- **XGBoost** [45]: An enhanced tree model that utilizes second-order gradient optimization, pruning, and regularization strategies to improve generalization and robustness. Source code is available at: `https://github.com/dmlc/xgboost`.

- **CatBoost** [46]: A boosting tree model optimized for categorical features. It employs an ordered boosting strategy to reduce prediction bias and is applicable to a wide range of structured data modeling tasks. Source code is available at: `https://github.com/catboost/catboost`.

- **DoubleEnsemble** [47]: Integrates multiple heterogeneous models and combines sample reweighting with feature selection mechanisms to enhance accuracy and robustness. Source code is available at: `https://github.com/microsoft/qlib/tree/main/examples/benchmarks/DoubleEnsemble`.

**Deep Learning Models.**

⇛ **General Deep Learning Models.**

- **Transformer** [49]: Utilizes multi-head self-attention mechanisms to capture long-range dependencies in time series data. It processes the entire sequence in parallel and offers better scalability compared to recurrent structures.

- **GRU** [22]: The Gated Recurrent Unit simplifies traditional recurrent neural networks by introducing update and reset gates, improving training efficiency and mitigating gradient vanishing.

- **LSTM** [23]: The Long Short-Term Memory network is a variant of recurrent neural networks, incorporating memory cells and gating mechanisms to model long-term dependencies effectively. It is one of the standard methods for time series modeling.

- **ALSTM** [48]: An enhanced version of the LSTM model that integrates an attention mechanism, enabling the model to focus on key time steps and selectively model sequence features.

- **GATs** [53]: Graph Attention Networks extend the attention mechanism to graph structures, enabling modeling of relationships among nodes in non-Euclidean space.

⇛ **Time-series Forecasting Models.**

- **PatchTST** [50]: A Transformer-based time series model that uses patching and channel independence techniques. It supports effective pretraining and transfer learning across datasets. Source code is available at: `https://github.com/yuqinie98/PatchTST`.

Table 5: Alpha 20 Baseline Factor Formulas

| Factor | Formula |
|---|---|
| RESI5 | $\text{Resi}(\$close, 5)/\$close$ |
| WVMA5 | $\text{Std}(|\$close/\text{Ref}(\$close, 1) - 1| \cdot \$volume, 5)/(\text{Mean}(|\$close/\text{Ref}(\$close, 1) - 1| \cdot \$volume, 5) + 1e^{-12})$ |
| RSQR5 | $\text{Rsquare}(\$close, 5)$ |
| KLEN | $(\$high - \$low)/\$open$ |
| RSQR10 | $\text{Rsquare}(\$close, 10)$ |
| CORR5 | $\text{Corr}(\$close, \log(\$volume + 1), 5)$ |
| CORD5 | $\text{Corr}(\$close/\text{Ref}(\$close, 1), \log(\$volume/\text{Ref}(\$volume, 1) + 1), 5)$ |
| CORR10 | $\text{Corr}(\$close, \log(\$volume + 1), 10)$ |
| ROC60 | $\text{Ref}(\$close, 60)/\$close$ |
| RESI10 | $\text{Resi}(\$close, 10)/\$close$ |
| VSTD5 | $\text{Std}(\$volume, 5)/(\$volume + 1e^{-12})$ |
| RSQR60 | $\text{Rsquare}(\$close, 60)$ |
| CORR60 | $\text{Corr}(\$close, \log(\$volume + 1), 60)$ |
| WVMA60 | $\text{Std}(|\$close/\text{Ref}(\$close, 1) - 1| \cdot \$volume, 60)/(\text{Mean}(|\$close/\text{Ref}(\$close, 1) - 1| \cdot \$volume, 60) + 1e^{-12})$ |
| STD5 | $\text{Std}(\$close, 5)/\$close$ |
| RSQR20 | $\text{Rsquare}(\$close, 20)$ |
| CORD60 | $\text{Corr}(\$close/\text{Ref}(\$close, 1), \log(\$volume/\text{Ref}(\$volume, 1) + 1), 60)$ |
| CORD10 | $\text{Corr}(\$close/\text{Ref}(\$close, 1), \log(\$volume/\text{Ref}(\$volume, 1) + 1), 10)$ |
| CORR20 | $\text{Corr}(\$close, \log(\$volume + 1), 20)$ |
| KLOW | $(\text{Less}(\$open, \$close) - \$low)/\$open$ |

- **iTransformer** [51]: A Transformer-based time series model that embeds each time series as variable tokens, improving parameter efficiency and modeling precision. It is suitable for long-sequence modeling tasks. Source code is available at: `https://github.com/thuml/iTransformer`.

- **Mamba** [52]: A next-generation long-sequence model based on state-space models, offering parallel computation and linear spatiotemporal complexity.

⇛ **Stock Prediction Models.**

- **TRA** [38]: Introduces a novel dynamic routing mechanism into the Transformer architecture, enabling the model to adaptively learn temporal patterns in stock prices and improve its ability to capture diverse market trends. Source code is available at: `https://github.com/TongjiFinLab/THGNN`.

- **MASTER** [39]: A market-centric Transformer model designed to dynamically model instantaneous and cross-temporal correlations among stocks, thereby improving trend prediction accuracy. Source code is available at: `https://github.com/SJTU-DMTai/MASTER`.

**Factor Libraries.**

- **Alpha 101** [41]: A collection of 101 formulaic trading alpha factors proposed by the WorldQuant team in 2015. Constructed using daily price-volume data, it represents an early publicly available benchmark of structured alpha factors in quantitative finance.

- **Alpha 158** [42]: Proposed by the Microsoft Qlib team, this library includes 158 traditional technical indicators (e.g., MA, RSI) constructed from combinations over different time windows (e.g., 5, 10, 20 days).

- **Alpha 360** [43]: A more comprehensive factor library provided by Microsoft Qlib, containing 360 factors constructed via normalization over historical price sequences (e.g., multi-period relative values of closing prices and volumes).

- **AutoAlpha** [44]: A dynamic structured factor library driven by large language models, integrating multimodal data such as text, numerical values, and images.

### C.4 Evaluation Details

### C.4.1 Metrics

We adopt two classes of metrics: factor-level predictive performance and strategy-level portfolio returns.

**Information Coefficient (IC).** IC measures the cross-sectional correlation between the predicted ranking and the realized return ranking. It is widely used in quantitative finance and defined as:

$$\text{IC} = \frac{(\hat{y} - \mathbb{E}[\hat{y}])^\top (y - \mathbb{E}[y])}{\sigma(\hat{y}) \cdot \sigma(y)} \tag{11}$$

where $\hat{y}$ and $y$ denote the predicted and realized rankings, respectively; $\mathbb{E}[\cdot]$ is the expectation, and $\sigma(\cdot)$ is the standard deviation. In practice, IC is computed daily and reported by its mean across time.

**Information Coefficient Information Ratio (ICIR).** ICIR evaluates the stability of IC over time and is defined as the ratio of the mean and standard deviation of daily IC values:

$$\text{ICIR} = \frac{\text{mean(IC)}}{\text{std(IC)}} \tag{12}$$

A higher ICIR indicates more consistent predictive ranking across trading days.

**Rank IC.** Rank IC refers to the Spearman rank correlation between the predicted and realized return rankings. It is robust to outliers and particularly suitable for distributions with heavy tails or extreme values.

**Rank ICIR.** Analogous to ICIR, Rank ICIR measures the temporal stability of Rank IC:

$$\text{Rank ICIR} = \frac{\text{mean(Rank IC)}}{\text{std(Rank IC)}} \tag{13}$$

It is a key indicator for assessing the long-term consistency of factor-based ranking models.

**Annual Return Ratio (ARR).** ARR reflects the compound annual growth rate of the portfolio:

$$\text{ARR} = \left( \prod_{t=1}^{T} (1 + r_t) \right)^{\frac{252}{T}} - 1 \tag{14}$$

where $r_t$ denotes the daily return, and $T$ is the total number of trading days.

**Information Ratio (IR).** IR evaluates the risk-adjusted excess return by comparing the annualized mean and standard deviation of returns relative to a benchmark:

$$\text{IR} = \frac{\text{mean}(r_t - r_b)}{\text{std}(r_t - r_b)} \times \sqrt{252} \tag{15}$$

where $r_b$ denotes the benchmark return (e.g., a market index or risk-free asset). When $r_b$ is set to the risk-free rate $r_f$, the IR coincides with the Sharpe Ratio.

In our setting, $r_b = r_f$, so IR and Sharpe Ratio values are numerically identical. To improve clarity, we denote this metric as IR (SHR*) in Table 1, Table 2, Table 7, and Table 8.

**Maximum Drawdown (MDD).** MDD measures the maximum loss from peak to trough during the evaluation period and captures downside risk:

$$\text{MDD} = \max_{t \in [1,T]} \left( \frac{\max_{i \in [1,t]} P_i - P_t}{\max_{i \in [1,t]} P_i} \right) \tag{16}$$

where $P_t$ is the portfolio value on day $t$, and $T$ is the evaluation horizon.

**Calmar Ratio.** The Calmar Ratio quantifies return relative to downside risk and is defined as:

$$\text{Calmar Ratio} = \frac{\text{ARR}}{|\text{MDD}|} \tag{17}$$

A higher Calmar Ratio indicates better return per unit of maximum loss, making it suitable for evaluating strategies that emphasize drawdown control.

### C.4.2 Trading Strategy

The full trading strategy is simulated as follows:

- On the close of trading day $t$, the model generates a ranking score for each stock based on its predicted return.

- At the open of trading day $t + 1$, the trader sells all holdings from day $t$ and selects the top 50 stocks by ranking to construct a new portfolio based on predicted returns. The bottom 5 performing stocks are excluded.

- Stocks that remain consistently highly ranked are retained in the portfolio to support the long-term holding of high-quality assets.

- During trade execution, a price limit threshold of 0.095 is applied. Trades are executed at the closing price, with a buy cost of 0.05%, a sell cost of 0.15%, and a minimum transaction fee of 5 CNY per trade.

## D  Supplementary Experiments

### D.1  Empirical Scope and Generalizability Analysis

To further evaluate the effectiveness and generalizability of `R&D-Agent(Q)`, we conducted a series of out-of-sample experiments on two additional markets, namely the CSI 500 and the NASDAQ 100.

For both datasets, we adopt a consistent temporal split, using the period from January 1, 2008, to December 31, 2021, for training, January 1, 2022, to December 31, 2023, for validation, and January 1, 2024, to June 30, 2025, for testing. Regarding the large language model (LLM) backends, we employ `GPT-4o`, whose training cutoff date is October 1, 2023, entirely preceding our test horizon, and `o4-mini`, whose training cutoff date is June 1, 2024, which remains almost entirely prior to the designated test period.

For the CSI 500 experiments, we apply the same trading settings as those used for the CSI 300 (see Appendix C.4.2). For the NASDAQ 100, we adopt market-specific settings, including portfolio rebalancing by selecting the top 20 stocks in each period (in contrast to 50 stocks for the CSI 300), a transaction cost of 0.1% per trade (as opposed to 0.5% in CSI 300), and the absence of daily price limits. The resulting out-of-sample performance on both the NASDAQ 100 and CSI 500 markets is summarized below.

As shown in Fig. 7 and Fig. 8, `R&D-Agent(Q)` consistently achieves strong out-of-sample performance across markets, instruments, and time periods not included in LLM training, providing further evidence for the robustness and real-world applicability of our approach.

### D.2  Ablation Analysis

Table 9 presents an extended ablation study of the `R&D-Agent(Q)` framework across two LLM backends (GPT-4o and o3-mini). We evaluate component-level contributions and compare three scheduling strategies for action selection: random, LLM-based, and contextual bandit.

**Component Ablation.** Removing the model branch (`R&D-Factor`) consistently yields stronger IC, ICIR, and ARR than removing the factor branch (`R&D-Model`). This reflects two effects: *(i)* factor optimization enables faster iteration and greater signal discovery under tight runtime; *(ii)* in early-stage pipelines, improved features have a larger impact than tuning the model. Nonetheless, `R&D-Model` contributes to portfolio-level risk smoothing (e.g., lower MDD with o3-mini).

**Algorithm Ablation.** For scheduling, random selection yields the lowest performance across both models. LLM-based decisions improve predictive quality but suffer from planning instability. The

Table 7: Out-of-sample experimental results on the CSI 500 dataset (tested from 2024 to June 2025), including factor predictive metrics and strategy performance metrics. Visual cues indicate ranking groups: Best , Second Best , Good (3–5) , Average (6–10) , Poor (11–15) , and Worse (16–19) .

| Models | | CSI500 | | | | | | | |
|---|---|---|---|---|---|---|---|---|---|
| | | Factor Predictive Power Metrics | | | | Performance Metrics | | | |
| | | IC | ICIR | Rank IC | Rank ICIR | ARR | IR (SHR*) | MDD | CR |
| Machine-Learning Models | LightGBM | 0.0181 | 0.1271 | 0.0393 | 0.2783 | -0.0294 | -0.3178 | -0.2089 | -0.1407 |
| | XGBoost | 0.0240 | 0.1675 | 0.0427 | 0.3054 | 0.0053 | 0.0634 | -0.1766 | 0.0300 |
| | CatBoost | 0.0241 | 0.1629 | 0.0390 | 0.2627 | 0.0111 | 0.1438 | -0.1799 | 0.0617 |
| | DoubleEnsemble | 0.0248 | 0.1705 | 0.0423 | 0.2850 | 0.0227 | 0.2500 | -0.2094 | 0.1084 |
| Deep-Learning Models | Transformer | 0.0194 | 0.1355 | 0.0416 | 0.2884 | 0.0234 | 0.2898 | -0.1331 | 0.1758 |
| | GRU | 0.0188 | 0.1022 | 0.0512 | 0.2711 | 0.0398 | 0.3716 | -0.1602 | 0.2484 |
| | LSTM | 0.0219 | 0.1434 | 0.0401 | 0.2825 | 0.0560 | 0.6900 | -0.1075 | 0.5209 |
| | GATs | 0.0162 | 0.1013 | 0.0426 | 0.2731 | 0.0478 | 0.5168 | -0.1569 | 0.3047 |
| | iTransformer | 0.0161 | 0.1031 | 0.0383 | 0.2278 | 0.0102 | 0.0985 | -0.1496 | 0.0682 |
| | TRA | 0.0260 | 0.1813 | 0.0464 | 0.3285 | 0.0504 | 0.6040 | -0.1461 | 0.3450 |
| Factor Libraries | Alpha 158 | 0.0192 | 0.1353 | 0.0374 | 0.2639 | 0.0199 | 0.2515 | -0.1771 | 0.1124 |
| | Alpha 360 | 0.0195 | 0.1331 | 0.0308 | 0.2089 | 0.0191 | 0.2527 | -0.1270 | 0.1504 |
| | AutoAlpha | 0.0184 | 0.1529 | 0.0175 | 0.1382 | 0.0397 | 0.5728 | -0.1006 | 0.3946 |
| R&D-Agent Series Framework* | R&D-Factor$_{GPT-4o}$ | 0.0201 | 0.1709 | 0.0176 | 0.1404 | 0.1010 | 1.3730 | -0.0787 | 1.2833 |
| | R&D-Factor$_{o4-mini}$ | 0.0264 | 0.2652 | 0.0345 | 0.3454 | 0.0849 | 1.0014 | -0.1215 | 0.6985 |
| | R&D-Model$_{GPT-4o}$ | 0.0259 | 0.1649 | 0.0532 | 0.3469 | 0.1039 | 1.0941 | -0.1367 | 0.7600 |
| | R&D-Model$_{o4-mini}$ | 0.0265 | 0.1825 | 0.0521 | 0.3616 | 0.1160 | 1.4021 | -0.0735 | 1.5777 |
| | R&D-Agent(Q)$_{GPT-4o}$ | 0.0241 | 0.1532 | 0.0555 | 0.3574 | 0.1358 | 1.4227 | -0.0803 | 1.6903 |
| | R&D-Agent(Q)$_{o4-mini}$ | 0.0288 | 0.1828 | 0.0564 | 0.3523 | 0.1982 | 2.1721 | -0.0656 | 3.0229 |

Table 8: Out-of-sample experimental results on the NASDAQ 100 dataset (tested from 2024 to June 2025), including factor predictive metrics and strategy performance metrics. Visual cues indicate ranking groups: Best , Second Best , Good (3–5) , Average (6–10) , Poor (11–15) , and Worse (16–19) .

| Models | | NASDAQ100 | | | | | | | |
|---|---|---|---|---|---|---|---|---|---|
| | | Factor Predictive Power Metrics | | | | Performance Metrics | | | |
| | | IC | ICIR | Rank IC | Rank ICIR | ARR | IR (SHR*) | MDD | CR |
| Machine-Learning Models | LightGBM | 0.0080 | 0.0652 | 0.0087 | 0.0842 | -0.0293 | -0.2603 | -0.1342 | -0.2183 |
| | XGBoost | 0.0076 | 0.0527 | 0.0112 | 0.0841 | 0.0169 | 0.1544 | -0.1211 | 0.1396 |
| | CatBoost | 0.0095 | 0.0614 | 0.0129 | 0.1005 | -0.0083 | -0.0735 | -0.1148 | -0.0723 |
| | DoubleEnsemble | 0.0047 | 0.0360 | 0.0086 | 0.0683 | -0.0005 | -0.0046 | -0.1404 | -0.0036 |
| Deep-Learning Models | Transformer | -0.0011 | -0.0077 | 0.0092 | 0.0686 | -0.0037 | -0.0343 | -0.1553 | -0.0238 |
| | GRU | 0.0064 | 0.0457 | 0.0147 | 0.1075 | 0.0347 | 0.2930 | -0.1504 | 0.2307 |
| | LSTM | 0.0062 | 0.0409 | 0.0150 | 0.1084 | 0.0550 | 0.4526 | -0.1204 | 0.4568 |
| | GATs | -0.0004 | -0.0023 | 0.0169 | 0.1015 | 0.0677 | 0.5772 | -0.1491 | 0.4541 |
| | iTransformer | 0.0076 | 0.0421 | 0.0041 | 0.0225 | 0.0617 | 0.3612 | -0.1991 | 0.3099 |
| | TRA | 0.0058 | 0.0446 | 0.0098 | 0.0825 | 0.0505 | 0.4608 | -0.1351 | 0.3738 |
| Factor Libraries | Alpha 158 | 0.0040 | 0.0324 | 0.0069 | 0.0624 | 0.0038 | 0.0303 | -0.1140 | 0.0333 |
| | Alpha 360 | 0.0042 | 0.0327 | 0.0086 | 0.0728 | 0.0756 | 0.5890 | -0.1182 | 0.6396 |
| | AutoAlpha | 0.0026 | 0.0265 | -0.0052 | -0.0432 | 0.0154 | 0.0974 | -0.1165 | 0.1326 |
| R&D-Agent Series Framework* | R&D-Factor$_{GPT-4o}$ | 0.0070 | 0.0446 | 0.0039 | 0.0357 | 0.1497 | 1.0985 | -0.0977 | 1.5335 |
| | R&D-Factor$_{o4-mini}$ | 0.0166 | 0.1017 | 0.0050 | 0.0407 | 0.1693 | 1.1169 | -0.0650 | 2.6059 |
| | R&D-Model$_{GPT-4o}$ | 0.0128 | 0.0831 | 0.0215 | 0.1427 | 0.1167 | 1.0742 | -0.0842 | 1.3869 |
| | R&D-Model$_{o4-mini}$ | 0.0081 | 0.0484 | 0.0213 | 0.1355 | 0.1367 | 1.2671 | -0.0741 | 1.8444 |
| | R&D-Agent(Q)$_{GPT-4o}$ | 0.0172 | 0.0908 | 0.0067 | 0.0490 | 0.2328 | 1.3312 | -0.1044 | 2.2292 |
| | R&D-Agent(Q)$_{o4-mini}$ | 0.0162 | 0.1035 | 0.0083 | 0.0673 | 0.2840 | 1.7737 | -0.0634 | 4.4814 |

Bandit scheduler consistently outperforms alternatives in IC, ARR, and valid loop count, demonstrating superior resource allocation by adapting to evolving performance signals.

Overall, the results highlight the factor branch as the main driver of signal quality, the model branch as a risk stabilizer, and the Bandit scheduler as an efficient mechanism to manage trade-offs under limited time and compute budgets.

### D.3 Factor Library Analysis

Fig. 10 shows the complete results of Factor Effects evaluation experiment. Beyond IC (subfigures (a) and (c)), subfigures (b) and (d) show consistent gains in Rank IC, confirming that R&D-Factor not only improves absolute prediction accuracy but also enhances relative ranking of stock returns.

Table 9: Ablation study of the `R&D-Agent(Q)` framework. The top rows show component-level ablations by disabling either factor or model generation. The bottom rows compare action selection strategies in `R&D-Agent(Q)`: random, LLM-based, and Bandit. Metrics include factor predictive power (IC, ICIR), strategy performance (ARR, MDD), and execution statistics (TL: total loops; VL: valid loops; SL: SOTA selections; TRH: total runtime in hours).

| Models | | Factor Predictive Power Metrics | | Performance Metrics | | Execution Metrics | | | |
|---|---|---|---|---|---|---|---|---|---|
| | | IC | ICIR | ARR | MDD | TL | VL | SL | TRH |
| GPT-4o | | | | | | | | | |
| Component Ablation | R&D-Factor | 0.0489 | 0.4050 | 0.1461 | -0.0750 | 36 | 33 | **9** | 6 |
| | R&D-Model | 0.0326 | 0.2305 | 0.1229 | -0.0876 | 23 | 12 | 5 | 6 |
| Algorithm Ablation | R&D-Agent(Q) w/ random | 0.0318 | 0.2431 | 0.0914 | -0.0782 | 36 | 18 | 7 | 12 |
| | R&D-Agent(Q) w/ LLM | **0.0523** | **0.4172** | 0.0940 | -0.0989 | 32 | 19 | 6 | 12 |
| | R&D-Agent(Q) w/ Bandit | 0.0497 | 0.4069 | **0.1144** | **-0.0811** | **38** | **22** | 8 | 12 |
| o3-mini | | | | | | | | | |
| Component Ablation | R&D-Factor | 0.0497 | 0.3931 | 0.1184 | -0.0910 | 28 | 16 | 6 | 6 |
| | R&D-Model | 0.0469 | 0.3688 | 0.1009 | -0.0694 | 30 | 15 | 7 | 6 |
| Algorithm Ablation | R&D-Agent(Q) w/ random | 0.0445 | 0.3589 | 0.0897 | -0.1004 | 33 | 19 | 7 | 12 |
| | R&D-Agent(Q) w/ LLM | 0.0476 | 0.3891 | 0.1009 | -0.0794 | 33 | 20 | 5 | 12 |
| | R&D-Agent(Q) w/ Bandit | **0.0532** | **0.4278** | **0.1421** | **-0.0742** | **44** | **24** | **8** | 12 |

Especially under Alpha 158 initialization, Rank IC remains above 0.07 in 2020 with `o3-mini`, while classical libraries decline sharply. This supports the claim that iterative refinement improves both signal strength and ranking consistency across regimes.

In terms of cumulative return (subfigure (e)), performance divergence becomes evident from early 2018. Factor sets generated by `R&D-Factor(158)` consistently outperform others, ending with a net asset value (NAV) exceeding 5.1 by 2020 Q3. Even `R&D-Factor(20)` configurations surpass Alpha360, indicating that larger factor sets do not necessarily yield higher returns. Traditional libraries suffer from increased volatility due to factor redundancy. In contrast, `R&D-Factor` mitigates this through dynamic filtering, achieving more stable and capital-efficient performance.

These results underscore `R&D-Factor`'s dual advantage in information efficiency (achieving higher IC/Rank IC with fewer factors) and capital efficiency (delivering superior NAV). Whether starting from a compact or high-dimensional base, its iterative refinement pipeline reliably discovers effective signals and removes redundancies, laying a solid foundation for subsequent model optimization and full-stack co-evolution in `RD-Quant`.

### D.4 Co-STEER Effectiveness Analysis

As a key component of the Development Phase in `R&D-Agent(Q)`, in addition to the direct implementation of `Co-STEER` within the `R&D-Agent(Q)` framework described in Section 4, we conducted further experiments to validate the capabilities of `Co-STEER`. Specifically, we want to answer the following research questions.

- **RQ1**: How well does `Co-STEER` perform in generating executable and semantically correct code for financial tasks, compared to recent code generation baselines?

- **RQ2**: Can its evolving scheduler improve implementation efficiency and output quality under constrained compute budgets?

**Dataset.** We evaluate `Co-STEER` on RD2Bench [76], a comprehensive benchmark for data-centric agent systems in finance. The benchmark encompasses both implementable 27 and non-implementable 13 factors, spanning fundamental, price-volume, and high-frequency categories. Each factor presents unique challenges, requiring sophisticated reasoning over heterogeneous financial data sources and the generation of executable Python code under strict constraints.

**Baselines.** We adopted Few-shot [71], CoT [35], Reflexion [72], Self-Debugging [74], and Self-Planning [73] as baselines. For details, see Section A.1.

**Metrics.** We introduced four evaluation metrics: average execution rate, average formatting correctness, average correlation, and maximum correlation. The average execution rate metric is used to measure the average success rate of code execution; any error encountered during execution is counted as 0. The average formatting correctness metric is used to measure the degree to which the generated code adheres to the correct format, such as whether column names meet expectations. The average

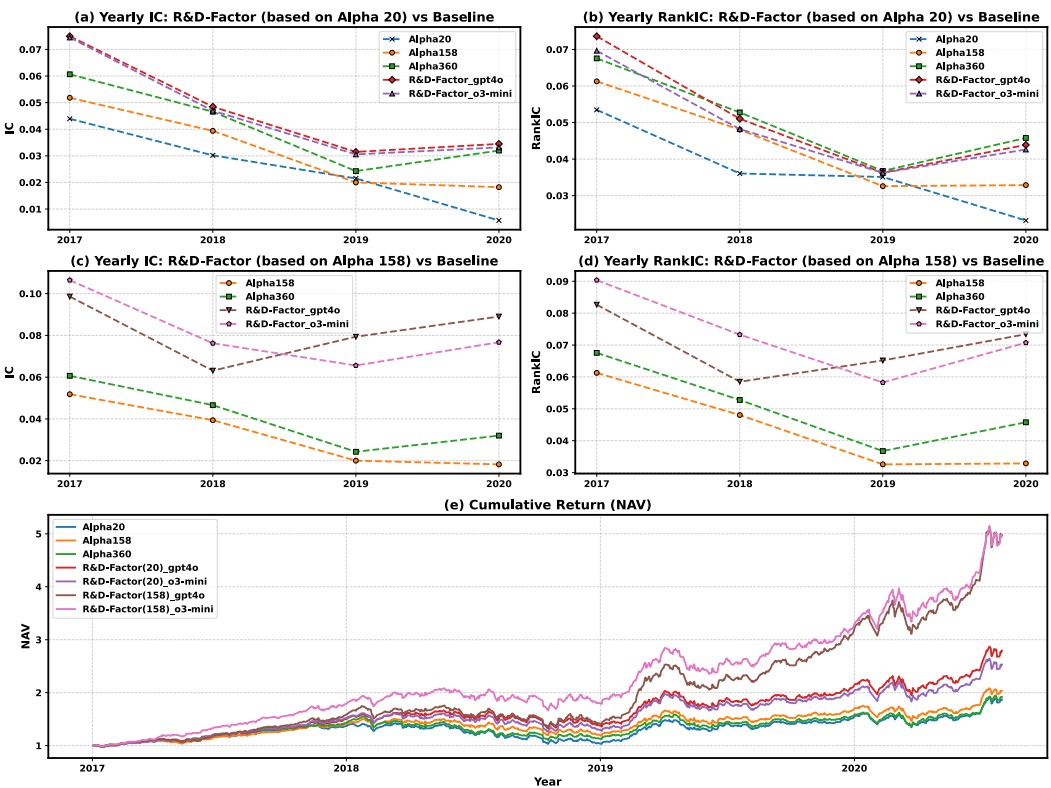

Figure 10: Comparison between classical factor libraries and `R&D-Factor`-generated factors using a LightGBM predictor on CSI 300. `R&D-Factor` was initialized from Alpha 20 or Alpha 158 and operated with `GPT-4o` or `o3-mini`. The top left figure shows the IC values of each method across different years, while the top right figure shows the RankIC values—the higher the value, the stronger the predictive power. The bottom figure presents the cumulative returns (NAV) of the corresponding strategies.

correlation metric reflects the average correlation between the code output sequence generated by the model and the ground truth results. For example, given the same input features, it evaluates the correlation between the factors generated by a large language model and those generated by actual implementation. The maximum correlation metric represents the highest correlation between the code output sequence generated by the model and the ground truth results.

**Results of Method Implementation (RQ1).**

Table 10: Results of method implementation. All the agent workflows are based on `GPT-4-turbo`.

| Methods | avg. exec. | avg. format | avg. corr. | max. corr. |
|---|---|---|---|---|
| Few-shot [71] | 0.733 | 0.433 | 0.454 | 0.562 |
| CoT [35] | 0.833 | 0.433 | 0.336 | 0.538 |
| Reflexion [72] | 0.822 | 0.400 | 0.269 | 0.550 |
| Self-Debugging [74] | 0.367 | 0.256 | 0.232 | 0.539 |
| Self-Planning [73] | 0.578 | 0.211 | 0.119 | 0.341 |
| Co-STEER (ours) | **0.889** | **0.611** | **0.646** | **0.887** |

Experimental results in Table 10 demonstrate `Co-STEER`'s superior implementation capabilities across all evaluation metrics on our 27 test cases. This performance advantage stems from two key innovations: dynamic knowledge expansion and contextual retrieval. While both Reflexion and Self-Debugging leverage environmental feedback (Table 4), `Co-STEER` uniquely accumulates and retrieves practical experience across implementations. Unlike existing approaches that only consider immediate feedback, `Co-STEER` builds a comprehensive knowledge base through continuous practice,

effectively bridging the expertise gap between junior and senior engineers. This systematic knowledge accumulation and retrieval mechanism enables `Co-STEER` to achieve significant performance gains across diverse implementation scenarios.

**Overall Performance Analysis (RQ2).** We evaluate `Co-STEER`'s effectiveness in a resource-constrained environment, where agents must optimize performance across 40 candidate methods (27 implementable, 13 non-implementable) with limited implementation attempts. This setup mirrors real-world computational constraints and tests the synergy between scheduling and implementation capabilities. Table 11 presents comparative results, revealing several key insights about system performance under practical constraints.

Table 11: Comparison of `Co-STEER` with random and evolving schedulers under top-$k$ evaluation (for $k = 5, 10, 15, 20$). Metrics include execution success rate (exec.), format correctness, and correlation with ground truth (average and maximum).

| Methods | Top 5 | | | | Top 10 | | | |
|---|---|---|---|---|---|---|---|---|
| | exec. | format | avg. corr. | max. corr. | exec. | format | avg. corr. | max. corr. |
| Random Scheduler | 0.522 | 0.400 | 0.211 | 0.444 | 0.567 | 0.289 | 0.417 | 0.655 |
| Evolving Scheduler | 0.765 | 0.259 | 0.280 | 0.515 | 0.815 | 0.358 | 0.519 | 0.778 |
| | **Top 15** | | | | **Top 20** | | | |
| Random Scheduler | 0.856 | 0.544 | 0.594 | 0.778 | 0.911 | 0.589 | 0.532 | 0.778 |
| Evolving Scheduler | 0.856 | 0.556 | 0.584 | 0.872 | **0.878** | **0.567** | **0.792** | **0.987** |

❶ **Evolving scheduling improves task effectiveness.** Table 11 shows that the evolving scheduler consistently outperforms the random baseline across all top-$k$ thresholds. This highlights its ability to learn effective execution orderings by identifying task complexity and dependencies. By accumulating experience over time, the system builds a form of engineering intuition, allowing it to prioritize easier or foundational tasks that unlock downstream implementation success.

❷ **More resources lead to stronger generalization.** As more budget is allocated, both schedulers benefit, but the evolving strategy shows greater gains. Unlike self-correction approaches that plateau early, the evolutionary process continues improving by retrieving and refining past attempts—regardless of whether initial trials were successful. This co-evolution between scheduler and implementation enables efficient adaptation under practical constraints.

### D.5 Cost Efficiency Analysis

Fig. 11 compares token expenditures of `GPT-4o` and `o3-mini` under fixed runtime settings. Factor tasks incur higher cost per loop due to their multi-stage structure—spanning hypothesis generation, implementation, and analysis for multiple candidates—whereas model tasks are simpler and less costly, as each loop generates only one model. `GPT-4o` and `o3-mini` show similar per-loop costs in model and quant settings. The larger gap in factor tasks stems from `o3-mini` generating more complex hypotheses per loop (producing more diverse factor types and handling more difficult implementations), resulting in higher costs. Despite these differences, both backends keep total costs under $10 across all `R&D-Agent(Q)` workflows (see Appendix C.1), confirming the framework's cost-effectiveness for scalable, automated quantitative research.

### D.6 Real-World Quantitative Competition Analysis

To further explore the potentials of `R&D-Agent(Q)` frameworks, we utilize `R&D-Agent(Q)` for the Optiver Realized Volatility Prediction [77] competition on `Kaggle`. This is a forecasting competition focused on predicting short-term volatility for hundreds of stocks using highly granular financial data. The competition challenges participants to predict the realized volatility of a set of stocks using information collected over a 10-minute time window, involving working with classic tabular time-series data and optimizing the Root Mean Squared Percentage Error (RMSPE) metric.

As shown in Fig. 12, the `R&D-Agent(Q)` achieved its best performance in the 12th experiment. According to the experiment summary, this experiment was based on the hypothesis that *by capturing the temporal evolution of bid-ask spreads across different time windows, the model's ability to predict short-term stock volatility can be enhanced.* The specific implementation involved calculating the

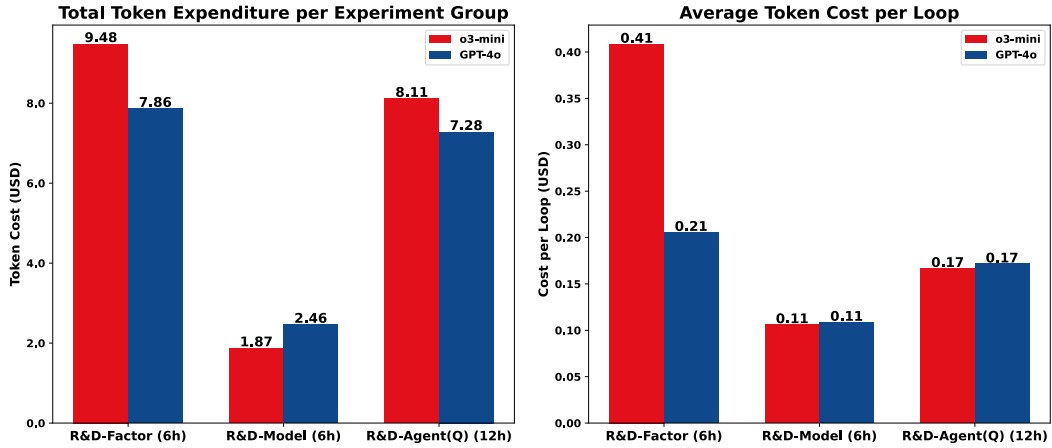

Figure 11: Token cost across different large language model backends (averaged over 5 trials). Left: total cost over corresponding runtimes (6h for `R&D-Factor`/`R&D-Model`, 12h for `R&D-Agent(Q)`); right: average per-loop cost. All costs are in USD.

rolling averages and standard deviations of bid-ask spreads over multiple time windows (5 seconds, 10 seconds, and 30 seconds) to efficiently capture the dynamic characteristics of market microstructure. Overall, from the optimization of the model in the 3rd round to the factor tuning in the 8th and 12th rounds, through continuous experimentation and exploration, `R&D-Agent(Q)` discovered that capturing the temporal features of bid-ask spread dynamics was most effective for this financial task. This also demonstrates that `R&D-Agent(Q)` can explore among many possible modeling approaches and, through empirical evaluation rather than relying solely on intuition or predetermined strategies, rationally identify the most promising directions. Furthermore, the `R&D-Agent(Q)` framework can be adapted to various quantitative financial tasks and performs relatively well.

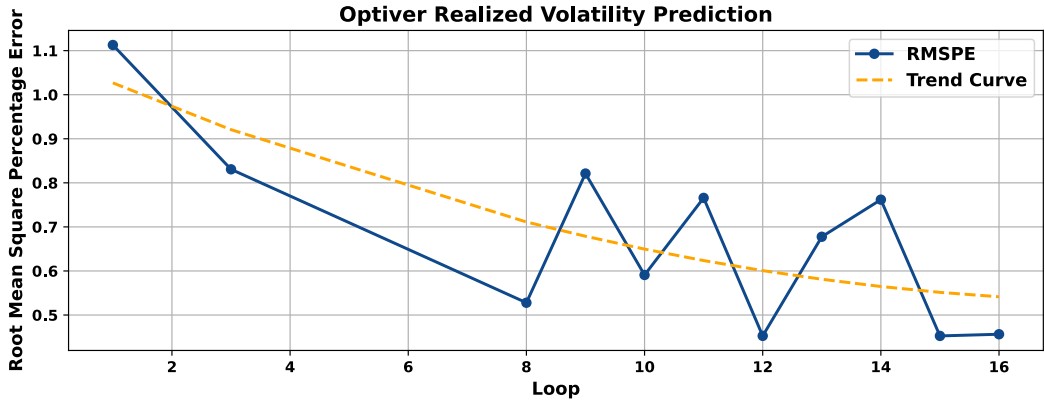

Figure 12: The RMSPE curve and fitted curve of `R&D-Agent` on Optiver Realized Volatility Prediction.

| Generated Hypotheses of `R&D-Agent` on Optiver Realized Volatility Prediction Competition |
|---|

**Experiment Loop Index 1**
**Action**: Construct Pipeline
**Hypothesis**: /
**Conclusion**: /

**Experiment Loop Index 2**
**Action**: Model
**Hypothesis**: Introducing a Gradient Boosting model, like LightGBM, with optimized hyper-

parameters will outperform the current linear regression model and ensemble strategy.
**Conclusion**: Failed.

**Experiment Loop Index 3**
**Action**: Model
**Hypothesis**: Replacing the linear regression model with a LightGBM model and using early stopping will improve the performance by capturing complex patterns better.
**Conclusion**: Success

...
**Experiment Loop Index 8**
**Action**: Factor
**Hypothesis**: Incorporating features that capture the interaction between bid and ask prices, such as bid-ask spread and order imbalance, will enhance the model's ability to predict short-term stock volatility.
**Conclusion**: Success

...
**Experiment Loop Index 12**
**Action**: Factor
**Hypothesis**: Incorporating features that capture the temporal evolution of bid-ask spreads over different time windows will enhance the model's ability to predict short-term stock volatility.
**Conclusion**: Success

...

# E   Prompt Design

## E.1   Specification Unit

As described in Section 2.1, the Specification Unit is responsible for dynamically generating the tuple $\mathcal{S}$ based on the current optimization objective. Downstream units selectively access components of $\mathcal{S}$ according to their functional roles—for instance, the Synthesis Unit and Analysis Unit typically use $\mathcal{B}$, $\mathcal{D}$, and $\mathcal{M}$, while the Implementation Unit relies on $\mathcal{D}$, $\mathcal{F}$, and $\mathcal{M}$.

Below, we provide the complete specification tuple for both factor and model optimization settings.

---

**Specification Prompt – Factor-Oriented**

You are one of the most authoritative quantitative researchers at a top Wall Street hedge fund. I need your expertise to design and implement new factors or models to enhance investment returns.
This time, I need your help with the research and development of the **factor**.

**Scenario Background**
- **Name**: Factor name.
- **Description**: Explanation of the factor logic.
- **Formulation**: Mathematical expression.
- **Variables**: All used variables or intermediate functions.

Clearly list all **hyperparameters**, such as lookback periods, window sizes, etc. Each factor must produce one output using a static data source. Different parameterizations count as different factors.

**Source Dataset**
`daily_pv.h5`
- Type: HDF5
- Index: MultiIndex [datetime, instrument]
- Columns: `$open, $high, $low, $close, $volume, $factor, ...`

**Output Format**
- Python file executable via: `python {your_file_name}.py`
- Includes: import section, function section, and a `main` function named `calculate_{function_name}`.

---

- Called under: if \_\_name\_\_ == "\_\_main\_\_"
- Do not use `try-except`.
- Output: Save computed factor to `result.h5` as a pandas DataFrame with index `[datetime, instrument]` and one column named by the factor.

**Workflow Mechanism**
1. Qlib generates a feature table from the factor values.
2. Trains models (e.g., LightGBM, LSTM, GRU) to predict future returns.
3. Builds portfolio based on predicted returns.
4. Evaluates performance (return, Sharpe ratio, max drawdown, etc.).

---

**Specification Prompt – Model-Oriented**

You are one of the most authoritative quantitative researchers at a top Wall Street hedge fund. I need your expertise to design and implement new models to enhance investment returns. This time, I need your help with the research and development of the **model**.

**Scenario Background** The model is a machine learning or deep learning structure used in quantitative investment to predict the returns and risks of a portfolio or a single asset. Models are employed to generate forecasts based on historical data and extracted factors, forming the core of many quantitative investment strategies.

Each model takes factor values as input and predicts future returns. Models are defined with a fixed architecture and hyperparameters to ensure reproducibility and consistency.

Each model should include the following components:
- **Name**: The name of the model.
- **Description**: Explanation of the model logic and purpose.
- **Architecture**: The internal structure of the model (e.g., LSTM layers, MLP, decision trees).
- **Hyperparameters**: All hyperparameters related to model structure.
- **Training_hyperparameters**: The hyperparameters used in training (e.g., learning rate, batch size).
- **ModelType**: One of "Tabular" or "TimeSeries" to indicate the input format.

The model must output one predicted return value. Different sets of hyperparameters define different models.

**Source Dataset**
`daily_pv.h5`
- Type: HDF5
- Index: MultiIndex [datetime, instrument]
- Columns: $open, $high, $low, $close, $volume, $factor, ...

**Output Format**
- Python file named `model.py` containing a PyTorch model definition.
- Includes the following parts:
  - **Import section**: Import only necessary libraries (e.g., `torch`, `torch.nn`).
  - **Model class**: A subclass of `torch.nn.Module` implementing `__init__` and `forward`.
  - **Model interface**: A variable named `model_cls` must be assigned to the defined model class.
- The model must follow these interface constraints:
  - For Tabular input: input shape = (batch_size, num_features).
  - For TimeSeries input: input shape = (batch_size, num_timesteps, num_features).
  - Output shape must always be (batch_size, 1).
  - The model must only use current input tensor, no external data loading or saving.
  - No other arguments will be passed; model must accept either: (num_features) or (num_features, num_timesteps).
- Do **not** include any `try-except` blocks.
- Do **not** include any training, inference, or saving logic.

> • The user will import the model class via: `from model import model_cls`
>
> **Workflow Mechanism**
> 1. Qlib generates a feature table from the factor values.
> 2. Trains models (e.g., LightGBM, LSTM, GRU) to predict future returns.
> 3. Builds portfolio based on predicted returns.
> 4. Evaluates performance (return, Sharpe ratio, max drawdown, etc.).

## E.2  Synthesis Unit

After receiving the dynamically assembled specification tuple from the Specification Unit, the Synthesis Unit leverages its evolving knowledge forest to propose new hypothesis. This is then decomposed into actionable research tasks.

Below is the prompt used when the optimization target is a factor:

---

**Hypothesis Synthesis Prompt – Factor-Oriented**

**System prompt:**

**Background information ...** (Received from Specification Unit)

The user has proposed several hypotheses and conducted evaluations. Your task is to analyze these trials, identify why those labeled `true` were successful, and why those labeled `false` failed. Then propose how to improve — either by refining existing approaches or exploring a new one.
If a new hypothesis is already provided in feedback and you agree with it, you may reuse it. Otherwise, generate an improved one.

**Guidelines for Hypothesis Generation:**
1. Each generation should produce 1–5 factors. Balance simplicity and complexity; use all available financial data.
2. Start with simple, easy-to-implement factors. Avoid complex or combined factors at the beginning. Clearly explain their rationale.
3. Increase complexity gradually. Introduce advanced or combined factors only after simpler ones are validated.
4. If several iterations fail to outperform SOTA, restart with a new direction beginning from simple factors. Optimize a given factor type from simple to complex.
5. Record factors that surpass SOTA to prevent redundant implementation.

**Output Format (JSON Schema):**

```
{
  "action": "factor",
  "hypothesis": "The new hypothesis generated based on the
      information provided.",
  "reason": "Comprehensive explanation for the new hypothesis."
}
```

---

**User prompt:**

**The former hypotheses and the corresponding feedbacks are as follows:**

`Trial 1`
- **Action:** factor
- **Hypothesis:** Develop simple momentum-based and price-volume factors using daily price and volume data.
- **Reason:** Momentum and price-volume factors are simple yet effective for quant investment. They capture underlying trends and trading activity, which can be indicative of future returns. Testing these straightforward factors will provide a

baseline for performance and help identify potential opportunities for more complex factor development in subsequent iterations.

- **Specific Factors:**
  - `factor_name`: 10_day_momentum
    `factor_description`: [Momentum Factor] The momentum factor captures the tendency of stocks with positive recent performance to continue performing well in the near future. Specifically, this factor calculates the return over the past 10 days.
    `factor_formulation`: $MOM_{10} = \frac{P_t}{P_{t-10}} - 1$
  - `factor_name`: 10_day_volatility
    `factor_description`: [Price-Volume Factor] The average volume factor captures the average trading volume over the past 10 days, indicating the level of trading activity. Higher trading volume can signal stronger price movements.
    `factor_formulation`: $VOL_{10} = \text{std}(R_{t-i})$, $i = 0 \ldots 9$
    - ...
- **Observation:** The newly developed momentum-based and price-volume factors show promising results in the context of the given hypothesis. All implemented factors consistently contributed to a performance that surpassed the previous SOTA results. Specifically, improvements were observed in terms of both the Information Coefficient (IC) and annualized return, which are critical metrics for assessing a predictive model's effectiveness. However, it is noted that the maximum drawdown has worsened compared to the SOTA benchmark.
- **Evaluation:** The results support the hypothesis that simple momentum-based and price-volume factors can enhance model performance in quantitative investment. The significant improvement in IC and annualized return suggests that these factors effectively capture underlying patterns and trends in stock performance.
- **Decision:** `True`

```
Trial 2
...
```

**The SOTA hypothesis and the corresponding feedback are as follows:**

- **Action:** factor
- **Hypothesis:** ...
- **Reason:** ...
- **Specific Factors:** ...
- **Observation:** ...
- **Evaluation:** ...
- **Decision:** `True`

**Last hypothesis and the corresponding feedback are as follows:**

- **Action:** factor
- **Hypothesis:** ...
- **Reason:** ...
- **Specific Factors:** ...
- **Observation:** ...
- **Evaluation:** ...
- **Decision:** ...
- **New Hypothesis (from the Analysis Unit, for reference):** ...
- **Reason:** ...

---

**Example Output:**

```
{
  "action": "factor",
  "hypothesis": "Incorporate advanced variations and
      combinations of existing momentum-based, price-volume, and
      volatility factors. Introduce factors like cumulative
      returns, turnover ratios, or volatility clustering
```

```
       measures to further refine performance while potentially
       minimizing drawdowns.",
    "reason": "The previous trials successfully demonstrated that
       basic momentum-based and price-volume factors can improve
       performance metrics like IC and annualized return.
       However, the increased drawdown indicates a potential need
       for more sophisticated risk control. By using advanced
       variations, such as factor combinations and new risk
       measures, we may optimize and address volatility
       concerns. This aligns with the feedback suggesting more
       complex factor formulations for enhanced predictability
       and stability in returns."
}
```

## Task Synthesis Prompt – Factor-Oriented

### System prompt:

The user is trying to generate new factors based on the hypothesis generated in the previous step. The factors are used in certain scenario, the scenario is as follows:

**Background information ...** (Received from Specification Unit)

The user will use the factors generated to do some experiments. The user will provide this information to you:
1. The target hypothesis you are targeting to generate factors for.
2. The hypothesis generated in the previous steps and their corresponding feedbacks.
3. Former proposed factors on similar hypothesis.
4. Some additional information to help you generate new factors.

### Output Format (JSON Schema):

```
{
  "factor name 1": {
    "description": "description of factor 1, start with its
        type, e.g. [Momentum Factor]",
    "formulation": "latex formulation of factor 1",
    "variables": {
      "variable or function name 1": "description of variable
          or function 1",
      "variable or function name 2": "description of variable
          or function 2"
    }
  },
  "factor name 2": {
    "description": "description of factor 2, start with its
        type, e.g. [Machine Learning based Factor]",
    "formulation": "latex formulation of factor 2",
    "variables": {
      "variable or function name 1": "description of variable
          or function 1",
      "variable or function name 2": "description of variable
          or function 2"
    }
  }
}
```

### User prompt:

The user has made several hypothesis on this scenario and did several evaluation on them.
**The target hypothesis you are targeting to generate factors for is as follows:**
**Chosen Action:** factor
**Hypothesis:** Develop simple momentum-based and price-volume factors using the daily price and volume data available.
**Reason:** Momentum and price-volume factors are simple yet effective for quant investment. They capture underlying trends and trading activity, which can be indicative of future returns. Testing these straightforward factors will provide a baseline for performance and help identify potential opportunities for more complex factor development in subsequent iterations.

The former hypothesis and the corresponding feedbacks are as follows:
...

---

**Example Output:**

```
{
  "cumulative_return_30_days": {
    "description": "[Momentum Factor] This factor measures the
        cumulative return of a stock over the past 30 days. It
        extends previous momentum factors by capturing a
        longer-term trend of price movements.",
    "formulation": "CR_{30} = \\prod_{i=0}^{29} (1 + R_{t-i}) -
        1",
    "variables": {
      "R_{t-i}": "Daily return at time t-i, defined as (P_{t-i}
        - P_{t-i-1}) / P_{t-i-1}"
    }
  },
  "turnover_ratio_20_days": {
    "description": "[Price-Volume Factor] This factor computes
        the average daily turnover ratio over the past 20 days,
        representing the liquidity and trading activity of the
        stock.",
    "formulation": "TR_{20} = \\frac{1}{20} \\sum_{i=0}^{19}
        \\frac{V_{t-i}}{\\text{Shares Outstanding}}",
    "variables": {
      "V_{t-i}": "Trading volume at time t-i",
      "Shares Outstanding": "Total number of shares outstanding
        for the stock"
    }
  }
  ...
}
```

### E.3 Implementation Unit

The Implementation Unit executes tasks proposed by the Synthesis Unit using the Co-STEER framework, which transforms high-level descriptions into executable code through iterative refinement. This process involves three key prompt stages: *(i)* Code Synthesis: Initial code generation based on task descriptions; *(ii)* Log Analysis: Parsing error traces or outputs to diagnose issues; *(iii)* Correctness Verification: Determining if the current code meets specification; if not, prompting for revision.

Each stage contributes to a self-correcting loop that enables robust execution even under imperfect initial synthesis.

Implementation Prompt – Factor-Oriented – Code Implementation

**System prompt:**

The user is trying to implement some factors in the following scenario:

**Background information ...** (Received from Specification Unit)

Your code is expected to align the scenario in any form which means the user needs to get the exact factor values with your code as expected.
To help you write the correct code, the user might provide multiple information that helps you write the correct code:
1. The user might provide you the correct code to similar factors. You should learn from these code to write the correct code.
2. The user might provide you the failed former code and the corresponding feedback to the code. The feedback contains the execution, the code and the factor value. You should analyze the feedback and try to correct the latest code.
3. The user might provide you the suggestion to the latest fail code and some similar fail-to-correct pairs. Each pair contains the fail code with similar error and the corresponding corrected version code. You should learn from these suggestions to write the correct code.
You must write your code based on your former latest attempt below which consists of your former code and code feedback. You should read the former attempt carefully and must not modify the correct parts of your former code.

**Output Format (JSON Schema):**

```
{
    "code": "The Python code as a string."
}
```

---

**User prompt:**

—— Target factor information: ——
factor_name: ...
factor_description: ...
factor_formulation: ...

[NOTE]
1. Ensure the computations are efficient. Prefer vectorized operations where possible, and consider JIT compilation (e.g., via numba) for recursive calculations if necessary.
2. Parallelization techniques (e.g., Joblib, Dask) are allowed if it improves performance.

**Here are some success implements of similar component tasks, take them as references:**
=====Factor 1:=====
factor_name: ...
factor_description: ...
factor_formulation: ...
=====Code:=====

```
# File Path: factor.py

```

=====Factor 2:=====
...

**Here are some wrong implements of similar component tasks, take them as references:**
=====Factor 1:=====
factor_name: ...
factor_description: ...
factor_formulation: ...
=====Code:=====

```
# File Path: factor.py

```

=====Factor 2:=====
...

---

---

**Example Output:**

```
{
    "code": "import pandas as pd\nimport numpy as np\n\n\ndef
        cumulative_return_30_days():\n..."
}
```

**System prompt:**

The User will provide you the information of the factor.

Your job is to check whether user's code is aligned with the factor and the scenario.

The user will provide the source Python code and the execution error message if execution failed.

The user might provide you the ground truth code for you to provide the critic. You should not leak the ground truth code to the user in any form, but you can use it to provide the critic. User has also compared the factor values calculated by the user's code and the ground truth code. The user will provide you some analysis result comparing the two outputs. You may find some error in the code which caused the difference between the two outputs.

If the ground truth code is provided, your critic should only consider checking whether the user's code is aligned with the ground truth code, since the ground truth is definitely correct.

If the ground truth code is not provided, your critic should consider checking whether the user's code is reasonable and correct.

Notice that your critics are not for user to debug the code. They are sent to the coding agent to correct the code. So do not give any following items for the user to check like "Please check the code line XXX."

Your suggestion should not include any code, just some clear and short suggestions. Please point out very critical issues in your response, ignore non-important issues to avoid confusion. If no big issue found in the code, you can respond "No critics found."

You should provide the suggestion to each of your critic to help the user improve the code. Please respond the critic in the following format. Here is an example structure for the output:

```
critic 1: The critic message to critic 1
critic 2: The critic message to critic 2
```

---

**User prompt:**

=====Factor information:=====
factor_name: ...
factor_description: ...
factor_formulation: ...

=====Python code:=====
*# File Path: factor.py*
``

=====Execution feedback:=====
Execution succeeded without error.
Expected output file found.
=====Factor value feedback:=====
The source dataframe has only one column which is correct.
The source dataframe does not have any infinite values.
The output format is correct. The dataframe has a MultiIndex with 'datetime' and 'instrument', a single column for the factor name, and the factor values are of type float32, which is

acceptable. The result aligns with the requirements.
The generated dataframe is daily.

---

---

## Example Output1:

```
No critics found.
```

## Example Output2:

```
critic 1: The main error arises due to the incorrect index
    handling when applying the rolling function over the grouped
    data. Specifically, ...
critic 2: The logic for handling empty or zero sums of volumes
    is present, but the application logic in rolling apply needs
    consistent indexing that matches the function's expectations.
```

## System prompt:

The user has finished evaluation and got some feedback from the evaluator.
The evaluator ran the code and obtained the factor value dataframe and provided several feedback items regarding the user's code and output. You should analyze the feedback and, considering the scenario and factor description, give a final decision about the evaluation result. The final decision concludes whether the factor is implemented correctly and, if not, detailed feedback containing the reason and suggestion if the final decision is `False`.

**The implementation final decision is considered in the following logic:**
1. If the value and the ground truth value are exactly the same under a small tolerance, the implementation is considered correct.
2. If the value and the ground truth value have a high correlation on IC or rank IC, the implementation is considered correct.
3. If no ground truth value is provided, the implementation is considered correct if the code executes successfully (assuming the data provided is correct). Any exceptions, including those actively raised, are considered faults of the code. Additionally, the code feedback must align with the scenario and factor description.

**Output Format (JSON Schema):**

```
{
    "final_decision": true,
    "final_feedback": "The final feedback message"
}
```

---

## User prompt:

=====Factor information:=====
factor_name: ...
factor_description: ...
factor_formulation: ...

=====Python code:=====

```
# File Path: factor.py

```

=====Execution feedback:=====
Execution succeeded without error.
Expected output file found.
=====Factor value feedback:=====
The source dataframe has only one column which is correct.

The source dataframe does not have any infinite values.
The output format is correct. The dataframe has a MultiIndex with 'datetime' and
'instrument', a single column for the factor name, and the factor values are of type float32,
which is acceptable. The result aligns with the requirements.
The generated dataframe is daily.

**Example Output:**

```
{
    "final_decision": "True",
    "final_feedback": "The factor '10_day_momentum' has been
        successfully implemented. The code executed without
        errors, and the resultant dataframe adheres to the
        specified requirements. The factor values are stored in
        the correct format and appear accurate given the nature
        of the momentum factor."
}
```

## E.4 Validation Unit

The Validation Unit does not involve any prompts.

## E.5 Analysis Unit

As described in Section 2.5, the Analysis Unit not only evaluates experimental outcomes but also
generates prompt-based feedback for local refinement. After each round, it uses structured prompts
to interpret the result triplet $(h^t, t^t, r^t)$, identify potential failure causes, and generate short-term
hypotheses targeting specific weaknesses (e.g., overfitting, poor generalization, feature instability).

These refined hypotheses are passed to the Synthesis Unit as context for the next generation cycle.
While the Analysis Unit operates on local, recent evidence, the Synthesis Unit integrates these
suggestions with global search memory—achieving a complementary balance between short-term
adaptation and long-term discovery.

---

**Analysis Prompt – Factor-Oriented**

**System prompt:**

You will receive a hypothesis, multiple tasks with their factors, their results, and the
SOTA result. Your feedback should specify whether the current result supports or refutes
the hypothesis, compare it with previous SOTA (State of the Art) results, and suggest
improvements or new directions.

Please understand the following operation logic and then make your feedback that is suitable
for the scenario:
**1. Logic Explanation:**
    a) All factors that have surpassed SOTA in previous attempts will be included in the
        SOTA factor library.
    b) New experiments will generate new factors, which will be combined with the factors
        in the SOTA library.
    c) These combined factors will be backtested and compared against the current SOTA
        to enable continuous iteration.
**2. Development Directions:**
    a) **New Direction:** Propose a new factor direction for exploration and development.
    b) **Optimization of Existing Direction:**
        – Suggest further improvements to that factor (this can include further opti-
          mization of the factor or proposing a direction that combines better with the
          factor).

       – Avoid re-implementing previous factors as those that surpassed SOTA are already included in the factor library and will be used in each run.

**3. Final Goal:** To continuously accumulate factors that surpass each iteration to maintain the best SOTA.

Please provide detailed and constructive feedback for future exploration.

**Output Format (JSON Schema):**

```
{
  "Observations": "Your overall observations here",
  "Feedback for Hypothesis": "Observations related to the
      hypothesis",
  "New Hypothesis": "Your new hypothesis here",
  "Reasoning": "Reasoning for the new hypothesis",
  "Replace Best Result": "yes or no"
}
```

---

User prompt:

**Target hypothesis:** Incorporate advanced variations and combinations of existing momentum-based, price-volume, and volatility factors. Introduce factors like cumulative returns, turnover ratios, or volatility clustering measures to further refine performance while potentially minimizing drawdowns.

**Tasks and Factors:**
- **cumulative_return_30_days:** [Momentum Factor] This factor measures the cumulative return of a stock over the past 30 days. It extends previous momentum factors by capturing a longer-term trend of price movements. **Implemented**.
- **turnover_ratio_20_days:** [Price-Volume Factor] This factor computes the average daily turnover ratio over the past 20 days, representing the liquidity and trading activity of the stock. **Not Implemented**.
- ...

**Combined Results:**
- IC: 0.033412 vs SOTA IC: 0.027691
- Annualized Return: 0.097140 vs SOTA: 0.076871
- Max Drawdown: -0.133149 vs SOTA: -0.075444

---

**Example Output:**

```
{
  "Observations": "The current results show a slight
      improvement in both the Information Coefficient (IC) and
      annualized return compared to the SOTA results. However,
      the max drawdown has increased slightly, indicating a
      potential increase in risk.",
  "Feedback for Hypothesis": "The hypothesis is partly
      supported as the introduction of advanced variations and
      combinations of factors has led to a slight improvement in
      the performance metrics, specifically the IC and
      annualized return. However, the hypothesis also aimed at
      potentially minimizing drawdowns, which was not achieved
      as the max drawdown increased.",
  "New Hypothesis": "Introduction of risk mitigation techniques
      alongside factor combinations could improve return metrics
      while effectively minimizing drawdowns.",
  "Reasoning": "While the advanced combinations have improved
      some aspects of the performance, the increased drawdown
      suggests the need to balance risk more effectively.
      Considering risk-adjusted factors or incorporating
      strategies such as variance reduction or diversification
```

```
        could further refine the improvements while maintaining or
        reducing drawdowns.",
    "Replace Best Result": "yes"
}
```

# F    Discussion

## F.1    Diagnostic Insight

The stability of automated quantitative research pipelines is often challenged by three scenarios: (i) noisy or sparse factors, (ii) exploration loops that fail to diversify, and (iii) sensitivity to the initial factor set. In our framework, each of these issues is explicitly considered.

1. **Noisy or sparse factors:** To prevent unreliable signals from propagating through iterations, Co-STEER integrates health-check modules that verify factors are leakage-free, non-trivial, and statistically meaningful within the training window. This mechanism filters out sparse or redundant candidates before they enter the optimization loop, as detailed in Section 2.4.

2. **Exploration inefficiency:** The multi-armed bandit scheduler is regularized by imposing an upper bound on the length of consecutive exploration in one direction. This prevents the system from getting trapped in a local loop and ensures adaptive balancing between factor-side and model-side refinements. Ablation studies in Appendix D.2 confirm that this mechanism improves both predictive quality and SOTA selections under limited resources.

3. **Initial factor sensitivity:** To reduce the dependence on starting conditions, generated factors are systematically deduplicated against the initial set, and the optimization process is isolated from raw definitions. Empirical results (Fig. 7, Fig. 10) show that when initialized with either small libraries (e.g., Alpha 20) or larger libraries (e.g., Alpha 158), the system converges toward diverse, high-quality factors. In particular, starting from Alpha 20, the framework rapidly achieves performance comparable to Alpha 158, while initialization from Alpha 158 yields further gains by building upon a stronger baseline. Moreover, out-of-sample evaluations across different markets (CSI 300, CSI 500, and NASDAQ 100) and time periods further demonstrate that these improvements are not restricted to specific datasets, but reflect consistent robustness and generalizability.

## F.2    Limitations and Future Works

While R&D-Agent(Q) shows compelling results in both real-world markets and quantitative competitions, we identify several limitations that outline clear paths for future research directions:

- **Multimodal Data Integration:** Although the system processes diverse market data, its factor generation could be enhanced by incorporating alternative data sources (e.g., news sentiment, macroeconomic indicators, and corporate filings) to capture richer market signals.

- **Domain Knowledge Incorporation:** While the current system already delivers strong results using general-purpose LLMs (e.g., GPT-4o, o3-mini), it relies solely on the models' built-in knowledge to propose financial hypotheses. Incorporating structured financial expertise—such as innovative solutions from financial reports or economic theory—through retrieval-augmented generation (RAG) could further enhance the plausibility, domain grounding, and efficiency of hypothesis generation.

- **Real-Time Market Adaptation:** The batch-based design restricts timely reactions to high-frequency trading. Incorporating event-driven or incremental learning could improve adaptability to regime shifts, anomalies, and emergent signals.

## F.3    Broader Impacts

R&D-Agent(Q) advances intelligent asset management through several transformative contributions:

- **Generalizable R&D Automation:** Although tailored to quantitative finance, our framework provides a modular, data-centric workflow that can be readily adapted to other scientific and engineering domains requiring hypothesis–implementation–validation cycles, potentially addressing bottlenecks in fields like healthcare, materials science, and operations research.

- **Reproducible and Deployable Outputs:** Every result produced by `R&D-Agent(Q)` is implemented as executable code. This design ensures end-to-end reproducibility and enables seamless deployment across different datasets or financial markets with minimal adaptation overhead.

- **Toward a New Financial AI Paradigm:** The framework unifies data-driven modeling and economic reasoning through a structured multi-agent design, offering a new foundation for interpretable, composable, and adaptive financial intelligence systems.

These advances position `R&D-Agent(Q)` as a foundational technology for the next decade of evidence-based quantitative investing.

While `R&D-Agent(Q)` lowers the barrier to building quantitative strategies, this accessibility also raises concerns that non-expert users may deploy generated factors or models directly in live trading without proper financial expertise or risk management. To mitigate this, we include clear disclaimers in the codebase stating that the framework is intended for research purposes only and that outputs require rigorous validation before real-world deployment.

### F.4 Large Language Model Usage

We use large language models (LLMs) as a core component of our framework—specifically, for the automated generation of trading factors and predictive models. All the settings of the LLM we used are provided in Appendix C.1. Apart from this, we only use the LLM for checking grammatical errors and formatting in the paper.

