# OpenReview forum: "R&D-Agent-Quant: A Multi-Agent Framework for Data-Centric Factors and Model Joint Optimization"
_NeurIPS.cc/2025/Datasets_and_Benchmarks_Track — NeurIPS 2025 Datasets and Benchmarks Track poster_

### Official Review · Reviewer_Sif6 · 2025-07-01

**Rating:** 5
**Confidence:** 3

**Summary:**

This paper introduces R&D-Agent(Q), a multi-agent system designed to automate the research and development process of quantitative trading strategies. The proposed framework incorporates modular agents that interact across five key R&D phases—Specification, Synthesis, Implementation, Validation, and Analysis—and jointly optimize both feature (factor) engineering and model selection. It also integrates Co-STEER, a structured code-generation agent, and a contextual Thompson Sampling mechanism to navigate the joint optimization space efficiently. While the problem is meaningful and the system design is compelling, several limitations related to framing, novelty, and evaluation prevent the paper from realizing its full potential in its current form.

**Additional Feedback:**

If the author can revise the paper or demonstrate a clear plan or potential to address all the aspects I mentioned above, I would be willing to upgrade my recommendation.

**Dataset Code Accessibility:**

Yes

**Ethical Considerations:**

No, there are no or only very minor ethics concerns

**Final Justification:**

The authors clearly answered the questions; please follow the suggested changes and citations, then I will upgrade the score from 3 to 5.

**Limitations Weaknesses:**

1. Limited novelty in learning methods: The techniques employed (bandits, code generation, standard backbones) are not novel individually. The contribution lies in system integration rather than algorithmic advancement, and this should be framed more modestly.

2. Narrow empirical scope: All experiments are conducted on CSI 300 stocks across 400 trading days. The absence of out-of-sample markets, instruments, or regimes limits generalizability. The Optiver example helps but is underdeveloped.

3. Lack of diagnostic insight: The paper would benefit from a discussion of when the framework struggles—e.g., with noisy factors, low signal-to-noise regimes, or sparse data. Does the bandit exploration get stuck? Is performance sensitive to initial factor setups?

4. Sharpe ratio as missing performance metric: While ARR and MDD are reported, the Sharpe ratio, which balances return and volatility, is not shown. Including it in the final result tables would offer a more complete picture of strategy quality, especially since MDD only captures a single extreme point and may not reflect general variability across the backtest.

6. Baseline selection could be broader: In addition to the models already compared, there are other factor-centric modeling approaches that should at least be acknowledged. For instance, “PolyModel for Hedge Funds’ Portfolio Construction Using Machine Learning” (Zhao et al.) offers a data-centric perspective that aligns with the paper’s theme. Even if not benchmarked, it deserves mention as part of the conceptual background.

**Strengths Contributions:**

1. Timely and relevant topic: Automating the quant strategy development cycle is a high-value and practically relevant problem. The idea of unifying data-centric factor discovery with model optimization is compelling, particularly in data-rich domains like finance.

2. Modular, extensible architecture: The division of agents into five modular phases—Specification through Analysis—mimics the real-world iterative R&D process. This lends itself well to transparency, extensibility, and interpretability.

3. Co-STEER for code generation: The integration of a structured code generation agent using retrieval-augmented prompting and multi-tool execution (e.g., LangChain, GitHub Copilot) is well-motivated. This helps bridge the gap between high-level research intent and low-level implementation.

4. Joint optimization via contextual bandits: The use of a contextual Thompson Sampling strategy to balance exploration across factor and model configurations is a thoughtful contribution. It offers a practical mechanism to manage the complexity of searching over a large joint space.

5. Solid empirical benchmarking: The experiments compare the framework against several well-established baselines on real-world stock data (CSI 300), and include a diverse set of metrics—IC, Rank IC, backtest PnL, and ablation results for each module.

---

> ### Author Rebuttal · Authors · 2025-07-31
>
> We sincerely thank the reviewer for the thorough and detailed feedback. Below are our responses.
>
> **For *W1* (Novelty and Framing):**
>
> > The contribution is mainly in system integration, as the individual techniques (bandits, code generation, standard backbones) are not themselves novel and should be framed more modestly.
>
> We agree with the reviewer that our key contribution is **not** in proposing entirely new algorithms, but in **systematically integrating** well-established techniques into a unified framework. Our biggest contribution is constructing the system with the most suitable techniques, which makes our system the first of its kind. To the best of our knowledge, $\text{RD-Agent(Q)}$ is the **first** end-to-end data-centric framework in quantitative finance that fully automates factor-model code R\&D through iterative refinement. We will highlight this main contribution more explicitly in the revision.
>
> In addition, our system does introduce several innovations within this integrated framework:
> - For standard backbones, we **pioneer the decoupling** of the quant research pipeline into five distinct stages specifically tailored for LLM agents. This structure helps address fundamental challenges such as *hallucination* and *poor interpretability*, supporting both transparency and extensibility.
> - For code generation, we propose the **Co-STEER Agent**, which innovatively combines systematic scheduling and adaptive code-generation strategies to ensure correctness and efficiency in iterative factor-model development cycles. Our experiments (Figures 6–8, Table 3) demonstrate its clear advantages over existing approaches.
>
> ---
> **For *W2* (Empirical Scope):**
>
> > All experiments are conducted on CSI 300 stocks across 400 trading days. The absence of out-of-sample markets, instruments, or regimes limits generalizability. The Optiver example helps but is underdeveloped.
>
> We appreciate the reviewer's insightful comment and for noting our experiments on the Optiver dataset (Appendix D.5).
>
> To further demonstrate the **effectiveness** and **generalizability** of our framework, we conducted additional out-of-sample experiments on both the **CSI 500** and **NASDAQ 100** markets.
>
> The data splits for both are as follows:
> * Train: 2008-01-01 to 2021-12-31
> * Validation: 2022-01-01 to 2023-12-31
> * Test: 2024-01-01 to 2025-06-30
>
> LLM Backend details:
> - `GPT-4o` (training cutoff: 2023-10-01, which is **completely** prior to our test period)
> - `o4-mini` (training cutoff: 2024-06-01, which is **almost entirely** prior to our test period)
>
> Trading setting notes:
> - For CSI 500: Settings are identical to CSI 300 (see Appendix C.4.2).
> - For NASDAQ 100:
>     - Portfolio rebalancing: Top 20 stocks selected each period (vs. 50 in CSI 300);
>     - Transaction cost: 0.1% per trade (vs. 0.5% in CSI 300);
>     - No daily price limit (unlike CSI 300).
>
> Below are the out-of-sample results on NASDAQ 100 and CSI 500:
>
> ### **Experiments on NASDAQ 100 and test from 2024 (Ori. Table 1)**
> |Model Type| Model|IC|ICIR|Rank IC|Rank ICIR|ARR|IR (SHR*)|MDD|CR|
> | :- | :- | :-: | :-: | :-: | :-: | :-: | :-: | :-:| :-: |
> |**Machine-Learning**|LightGBM|0.008|0.065|  0.009|0.084|-0.029|-0.260|-0.134|-0.218|
> ||XGBoost|0.008|0.053|0.011|0.084|0.017|0.154|-0.121|0.140|
> ||CatBoost|0.010|0.061|0.013|0.101|-0.008|-0.074|-0.115|-0.072|
> ||DoubleEnsemble|0.005|0.036|0.009|0.068|-0.001|-0.005|-0.140|-0.004|
> |**Deep-Learning**|Transformer|-0.001|-0.008|0.009|0.069|-0.004|-0.034| -0.155|-0.024|
> ||GRU|0.006|0.046|0.015|0.108|0.035|0.293|-0.150|0.231|
> ||LSTM|0.006|0.041|0.015|0.108|0.055|0.4523|-0.120|0.457|
> ||GATs|0.000|-0.002|0.017|0.102|0.068|0.577|-0.150|0.454|
> ||iTransformer|0.008|0.042|0.004|0.023|0.062|0.361|-0.199|0.310|
> ||TRA|0.006|0.045|0.010|0.083|0.051|0.461|-0.135|0.374|
> |**Factor Libraries**|Alpha 158|0.004|0.032|0.007|0.062|0.004|0.030|-0.114|0.033|
> ||Alpha 360|0.004|0.033|0.009|0.073|0.076|0.589|-0.118|0.640|
> ||AutoAlpha|0.005|0.027|-0.005|-0.043|0.015|0.097|-0.117|0.133|
> |**RD-Agent Series**|RD-Factor$_{\text{GPT-4o}}$|0.007|0.045|0.004|0.036|0.150|1.099|-0.0978|1.534|
> ||RD-Factor$_{\text{o4-mini}}$|$\underline{0.017}$|$\underline{0.102}$|0.005|0.041|0.169|1.117|$\underline{-0.065}$|$\underline{2.606}$|
> ||RD-Model$_{\text{GPT-4o}}$|0.013|0.083|**0.022**|**0.143**|0.117|1.074|-0.084|1.387|
> ||RD-Model$_{\text{o4-mini}}$|0.008|0.048|$\underline{0.021}$|$\underline{0.136}$|0.137|1.267|-0.074|1.844|
> ||RD-Agent(Q)$_{\text{GPT-4o}}$|**0.017**|0.091|0.007|0.049|$\underline{0.233}$|$\underline{1.331}$|-0.104|2.229|
> ||RD-Agent(Q)$_{\text{o4-mini}}$|0.016|**0.104**|0.008|0.067|**0.284**|**1.774**|**-0.063**|**4.481**|
> ### **Experiments on CSI 500 and test from 2024 (Ori. Table 1)**
> |Model Type| Model|IC|ICIR|Rank IC|Rank ICIR|ARR|IR (SHR*)|MDD|CR|
> | :- | :- | :-: | :-: | :-: | :-: | :-: | :-: | :-:| :-: |
> |**Machine-Learning**|LightGBM|0.018|0.127|0.039|0.278|-0.029|-0.318|-0.209|-0.141|
> ||XGBoost|0.024|0.168|0.043|0.305|0.005|0.063|-0.177|0.030|
> ||CatBoost|0.024|0.163|0.039|0.263|0.011|0.144|-0.180|0.062|
> ||DoubleEnsemble|0.025|0.171|0.042|0.285|0.023|0.250|-0.209|0.108|
> |**Deep-Learning**|Transformer|0.019|0.1365|0.042|0.288|0.023|0.290|-0.133|0.176|
> ||GRU|0.019|0.102|0.051|0.271|0.040|0.372|-0.160|0.248|
> ||LSTM|0.022|0.143|0.040|0.283|0.056|0.690|-0.108|0.521|
> ||GATs|0.016|0.101|0.043|0.273|0.048|0.517|-0.157|0.305|
> ||iTransformer|0.016|0.103|0.038|0.228|0.010|0.099|-0.150|0.068|
> ||TRA|0.026|0.181|0.046|0.329|0.050|0.604|-0.146|0.345|
> |**Factor Libraries**|Alpha 158|0.019|0.135|0.037|0.264|0.020|0.252|-0.177|0.112|
> ||Alpha 360|0.020|0.133|0.031|0.209|0.019|0.253|-0.127|0.150|
> ||AutoAlpha|0.018|0.153|0.018|0.138|0.040|0.573|-0.101|0.395|
> |**RD-Agent Series**|RD-Factor$_{\text{GPT-4o}}$|0.020|0.171|0.018|0.140|0.101|1.373|-0.079|1.283|
> ||RD-Factor$_{\text{o4-mini}}$|0.026|**0.265**|0.035|0.345|0.085|1.001|-0.122|0.699|
> ||RD-Model$_{\text{GPT-4o}}$|0.026|0.165|0.053|0.347|0.104|1.094|-0.137|0.760|
> ||RD-Model$_{\text{o4-mini}}$|$\underline{0.027}$|0.183|0.052|**0.362**|   0.116|1.402|$\underline{-0.074}$|1.578|
> ||RD-Agent(Q)$_{\text{GPT-4o}}$|0.024|0.153|$\underline{0.056}$|$\underline{0.357}$|$\underline{0.136}$|$\underline{1.423}$|-0.080|$\underline{1.690}$|
> ||RD‑Agent(Q)$_{\text{o4-mini}}$|**0.029**|$\underline{0.183}$|**0.056**|0.352|**0.198**|**2.172**|**-0.066**|**3.023**|
>
> As shown, $\text{RD-Agent(Q)}$ consistently achieves strong out-of-sample performance across **markets**, **instruments**, and **time periods not included in LLM training**, providing further evidence for the robustness and real-world applicability of our approach.
>
> ---
> **For *W3* (Diagnostic Insight):**
>
> > A discussion is needed on when the framework struggles, such as with noisy or sparse factors, low signal-to-noise regimes, potential issues in bandit exploration, and sensitivity to initial factor setups.
>
> We appreciate these insightful comments and could not agree more on the importance of the diagnostic challenges mentioned. All these challenges have been carefully addressed in $\text{RD-Agent(Q)}$:
>
> - For noisy or sparse factors, we implemented several **health check modules** in Co-STEER to ensure factors are safe (without leakage), usable (not sparse or identical), and robust (having signal in the training period).
> - For bandit exploration getting stuck, we set an upper threshold on how long the loop can continuously perform one type of exploration.
> - For initial factor setups, we isolate the agent from the details of the initial factor setup and perform rule-based deduplication between the generated factors and the initial factors.
>
> These details have been instrumental in enhancing the framework's robustness and stability. While previously omitted from the main text to avoid distracting from the core narrative, we recognize their importance and will include a discussion in the main body and provide implementation details in the appendix in the revision.
>
> ---
> **For *W4* (Performance Metrics):**
>
> > Sharpe ratio is missing, which balances return and volatility and offers a more complete assessment than ARR and MDD alone.
>
> According to Wikipedia, the Information Ratio (IR) and Sharpe Ratio are similar but slightly different. The IR is defined as:
> $$
> IR = \frac{R_p - R_b}{\sigma(R_p - R_b)}
> $$
> where $R_p$ is the portfolio return and $R_b$ is the benchmark return. By contrast, the Sharpe Ratio measures return relative to the risk-free rate $R_f$:
>
> $$
> SHR = \frac{R_p - R_f}{\sigma(R_p - R_f)}
> $$
>
> We appreciate this comment and fully agree on the importance of risk-adjusted performance metrics. In our setting, the benchmark return is set as a *risk-free asset*, so IR and Sharpe Ratio results are exactly **the same**. We will clarify this distinction in the revision, relabel **IR** in Table 1 to **IR (SHR\*)**, and add a footnote explaining the difference from the Sharpe Ratio to improve clarity.
>
> ---
> **For *W5* (Baseline Coverage):**
>
> > The baseline selection could be broader; additional factor-centric modeling approaches, such as Zhao et al.'s “PolyModel” should be acknowledged, even if not benchmarked, to provide a more comprehensive conceptual background.
>
> In fact, we provide not only model-level baseline comparisons but also extensive factor-level comparisons, including several classic factor models such as Alpha101, Alpha158, and AutoAlpha, to validate the effectiveness of our framework (see Table 1, Figures 7 and 10).
>
> We appreciate the reviewer's suggestion to consider additional factor-centric modeling approaches. After careful review, we note that while Zhao et al.'s work offers valuable insights into factor selection and portfolio optimization in hedge funds, its methodological focus and application domain differ from ours, as our approach is based on the data-driven generation of novel factors for portfolio optimization. Nonetheless, we will acknowledge this work in the related work section to provide broader context on factor-centric modeling in quantitative finance.

---

### Official Review · Reviewer_rHYN · 2025-07-02

**Rating:** 5
**Confidence:** 4

**Summary:**

This paper proposes RD-Agent-Quant, which is a large language model (LLM)-based framework that automates the discovery of factors that predict changes in the stock market. RD-Agent-Quant is inspired by the process of doing research by human beings. It gets LLM to create iterative prototypes of the model and does not require human intervention. It has been demonstrated to outperform prior factor-based frameworks on the Chinese stock market, as measured by a variety of quantitative finance performance metrics and factor predictive power metrics.

**Additional Feedback:**

Presentation areas for improvement:
1.	In Figure 1, it would be better to label the stages for easier reading. You can replace the figure with the figure in the following link: https://private-user-images.githubusercontent.com/465606/446348319-3198bc10-47ba-4ee0-8a8e-46d5ce44f45d.png?jwt=eyJhbGciOiJIUzI1NiIsInR5cCI6IkpXVCJ9.eyJpc3MiOiJnaXRodWIuY29tIiwiYXVkIjoicmF3LmdpdGh1YnVzZXJjb250ZW50LmNvbSIsImtleSI6ImtleTUiLCJleHAiOjE3NTEzODE1MDQsIm5iZiI6MTc1MTM4MTIwNCwicGF0aCI6Ii80NjU2MDYvNDQ2MzQ4MzE5LTMxOThiYzEwLTQ3YmEtNGVlMC04YThlLTQ2ZDVjZTQ0ZjQ1ZC5wbmc_WC1BbXotQWxnb3JpdGhtPUFXUzQtSE1BQy1TSEEyNTYmWC1BbXotQ3JlZGVudGlhbD1BS0lBVkNPRFlMU0E1M1BRSzRaQSUyRjIwMjUwNzAxJTJGdXMtZWFzdC0xJTJGczMlMkZhd3M0X3JlcXVlc3QmWC1BbXotRGF0ZT0yMDI1MDcwMVQxNDQ2NDRaJlgtQW16LUV4cGlyZXM9MzAwJlgtQW16LVNpZ25hdHVyZT04OWJlY2IzOWIyNWUzNjBjNmU5OTFmZWRkZGNjYzJjYTFmOTJlNjcyMmM4Y2MwNjJjOGI0MDY1ZGFiNDg1MDQwJlgtQW16LVNpZ25lZEhlYWRlcnM9aG9zdCJ9.F7votX0KcgANGin2M64K9lvy-Yy3Giso9gcHz3en4K4
2.	I would expect chain-of-thought to be cited in line 75.
3.	It would be best to explicitly mention in Section 2, the units that use LLM, and how LLM is applied. It was only until I read the prompting appendix when things get a little clearer. Otherwise, readers who read the paper without the appendix would be confused on how the units are implemented, and would have the following question on their head: ‘Is this achieved with supervised learning, reinforcement learning or LLM?’

**Dataset Code Accessibility:**

Yes

**Dataset Code Comments:**

Both the model code and the dataset are accessible. Locating the dataset code is tricky because I have to look at the correct section of the paper. Thankfully, the checklist makes it easier to locate the dataset code.

**Ethical Comments:**

NIL

**Ethical Considerations:**

No, there are no or only very minor ethics concerns

**Final Justification:**

Most of my concerns are solved, hence I will raise my score. The impact of the work is expected to be far-reaching in the world of quantitative finance, as large language models can now be applied to discover factors that can help improve stock market returns.

**Limitations Weaknesses:**

The evaluation is relatively limited because the evaluation is on Chinese stocks rather than the global stock market. If the authors can improve the evaluation, I will consider increasing the score.

weakness:
1.	Only Chinese stocks are considered. I cannot tell if the proposed framework only works well with Chinese stocks or can be generalized to the entire stock market.
2.	The evaluation testing time frame is only until 1 August 2020. It would be better if you could test on more recent data. Let me know if this is not possible.
3.	The pretrained LLM used has a knowledge cutoff later than the end of the testing set. This could lead to data leakage, with the benefit of hindsight from knowledge trained based on what actually happened in the testing set. But to be fair, this is hard to address with limited stock market data from the release of ChatGPT.
4.	In line 129, G is strictly speaking, not a mathematical function, because the temperature of the LLM is nonzero (from Appendix C.1), which implies that the output is not deterministic. This is because a mathematical function must have deterministic output.

**Strengths Contributions:**

The paper is novel and insightful. Impact is expected to be far-reaching, beyond quantitative finance. For instance, I expect that the method can be extended towards solving competitive programming problems, recommender systems and suggesting hypotheses for human beings to conduct future research.

Strengths:
1.	Simple and novel at the same time. The centre of the work is prompt engineering. Any person familiar with LLM can do effective prompt engineering, highlighting the simplicity of the approach. Prompt engineering, by itself, is not novel. However, the use of prompt engineering to do automated research on quantitative finance factors, without any human intervention, is novel.
2.	Impactful, with expected wide-ranging applications and extensions.
3.	Depth of analysis, extending to the appendix sections that contain the ablation studies.

---

> ### Author Rebuttal · Authors · 2025-07-31
>
> We sincerely thank the reviewer for the thorough and detailed feedback. Below are our responses.
>
> **For *Limitation* & *W1* (Empirical Scope and Generalizability):**
>
> > The evaluation is limited to Chinese stocks only, raising questions about the framework's generalizability beyond this market.
>
> This is a valuable suggestion. To address it, we conducted additional experiments on the U.S. stock market using the **NASDAQ 100 dataset**, split as follows:
>
> * Train: 2008-01-01 to 2021-12-31
> * Validation: 2022-01-01 to 2023-12-31
> * Test: 2024-01-01 to 2025-06-30
>
> Trading setting differences (relative to Table 1, CSI 300; see Appendix C.4.2):
> - Portfolio rebalancing: Top 20 stocks are selected each period (vs. top 50 in CSI 300)
> - Transaction cost: 0.1% per trade (vs. 0.5% in CSI 300)
> - No daily price limit in NASDAQ (unlike CSI 300)
>
> Below are the out-of-sample results on NASDAQ 100:
>
> ### **Experiments on NASDAQ 100 and test from 2024 (Ori. Table 1)**
> |Model Type| Model|IC|ICIR|Rank IC|Rank ICIR|ARR|IR (SHR\*)|MDD|CR|
> | :- | :- | :-: | :-: | :-: | :-: | :-: | :-: | :-:| :-: |
> |**Machine-Learning**|LightGBM|0.0080|0.0652|  0.0087|0.0842|-0.0293|-0.2603|-0.1342|-0.2183|
> ||XGBoost|0.0076|0.0527|0.0112|0.0841|0.0169|0.1544|-0.1211|0.1396|
> ||CatBoost|0.0095|0.0614|0.0129|0.1005|-0.0083|-0.0735|-0.1148|-0.0723|
> ||DoubleEnsemble|0.0047|0.0360|0.0086|0.0683|-0.0005|-0.0046|-0.1404|-0.0036|
> |**Deep-Learning**|Transformer|-0.0011|-0.0077|0.0092|0.0686|-0.0037|-0.0343| -0.1553|-0.0238|
> ||GRU|0.0064|0.0457|0.0147|0.1075|0.0347|0.2930|-0.1504|0.2307|
> ||LSTM|0.0062|0.0409|0.0150|0.1084|0.0550|0.4526|-0.1204|0.4568|
> ||GATs|-0.0004|-0.0023|0.0169|0.1015|0.0677|0.5772|-0.1491|0.4541|
> ||iTransformer|0.0076|0.0421|0.0041|0.0225|0.0617|0.3612|-0.1991|0.3099|
> ||TRA|0.0058|0.0446|0.0098|0.0825|0.0505|0.4608|-0.1351|0.3738|
> |**Factor Libraries**|Alpha 158|0.0040|0.0324|0.0069|0.0624|0.0038|0.0303|-0.1140|0.0333|
> ||Alpha 360|0.0042|0.0327|0.0086|0.0728|0.0756|0.5890|-0.1182|0.6396|
> ||AutoAlpha|0.0046|0.0265|-0.0052|-0.0432|0.0154|0.0974|-0.1165|0.1326|
> |**RD-Agent Series**|RD-Factor$_{\text{GPT-4o}}$|0.0070|0.0446|0.0039|0.0357|0.1497|1.0985|-0.0977|1.5335|
> ||RD-Factor$_{\text{o4-mini}}$|$\underline{0.0166}$|$\underline{0.1017}$|0.0050|0.0407|0.1693|1.1169|$\underline{-0.0650}$|$\underline{2.6059}$|
> ||RD-Model$_{\text{GPT-4o}}$|0.0128|0.0831|**0.0215**|**0.1427**|0.1167|1.0742|-0.0842|1.3869|
> ||RD-Model$_{\text{o4-mini}}$|0.0081|0.0484|$\underline{0.0213}$|$\underline{0.1355}$|0.1367|1.2671|-0.0741|1.8444|
> ||RD-Agent(Q)$_{\text{GPT-4o}}$|**0.0172**|0.0908|0.0067|0.0490|$\underline{0.2328}$|$\underline{1.3312}$|-0.1044|2.2292|
> ||RD-Agent(Q)$_{\text{o4-mini}}$|0.0162|**0.1035**|0.0083|0.0673|**0.2840**|**1.7737**|**-0.0634**|**4.4814**|
>
> As shown, our $\text{RD-Agent(Q)}$ series achieves the **best** overall results across multiple performance metrics, including excess return (ARR), risk-adjusted return (IR), and maximum drawdown (MDD), clearly outperforming established factor- and model-level baselines. These findings confirm that our framework generalizes well beyond the Chinese market and demonstrates robust, consistent performance in a **global** context.
>
> ---
>
> **For *W2* & *W3* (Recency and Data Leakage):**
>
> > The evaluation period ends on August 1, 2020, which may limit assessment of recent market conditions. Also, the pretrained LLM's knowledge cutoff is later than the test period, possibly causing data leakage due to hindsight bias.
>
> This is an important question. Our framework leverages a data-centric approach that provides distinct advantages. Specifically, we do not expose the LLM to real-time data or explicit dataset splits; rather, the LLM only receives the data schema and volume information, as shown in the prompt for the Specification Unit (Appendix E.1). This design prevents the LLM from knowing precise temporal boundaries or training-validation-test partitions, thereby effectively mitigating information leakage concerns.
>
> To further address this issue, we conducted additional experiments on the **CSI 500 dataset** with a clearly separated and more recent time split:
> * Train: 2008-01-01 to 2021-12-31
> * Validation: 2022-01-01 to 2023-12-31
> * Test: 2024-01-01 to 2025-06-30
>
> For LLM backends, we use
> - `GPT-4o` (training cutoff: 2023-10-01, which is **completely** prior to our test period)
> - `o4-mini` (training cutoff: 2024-06-01, which is **almost entirely** prior to our test period)
>
> *(Due to temporary API limitations, `o3-mini` was replaced by `o4-mini`, which has comparable capacity and performance.)*
>
> Below are the experimental results demonstrating the robustness of our method under this strict temporal partition:
>
> ### **Experiments on CSI 500 and test from 2024 (Ori. Table 1)**
> |Model Type| Model|IC|ICIR|Rank IC|Rank ICIR|ARR|IR (SHR\*)|MDD|CR|
> | :- | :- | :-: | :-: | :-: | :-: | :-: | :-: | :-:| :-: |
> |**Machine-Learning**|LightGBM|0.0181|0.1271|0.0393|0.2783|-0.0294|-0.3178|-0.2089|-0.1407|
> ||XGBoost|0.0240|0.1675|0.0427|0.3054|0.0053|0.0634|-0.1766|0.0300|
> ||CatBoost|0.0241|0.1629|0.0390|0.2627|0.0111|0.1438|-0.1799|0.0617|
> ||DoubleEnsemble|0.0248|0.1705|0.0423|0.2850|0.0227|0.2500|-0.2094|0.1084|
> |**Deep-Learning**|Transformer|0.0194|0.1355|0.0416|0.2884|0.0234|0.2898|-0.1331|0.1758|
> ||GRU|0.0188|0.1022|0.0512|0.2711|0.0398|0.3716|-0.1602|0.2484|
> ||LSTM|0.0219|0.1434|0.0401|0.2825|0.0560|0.6900|-0.1075|0.5209|
> ||GATs|0.0162|0.1013|0.0426|0.2731|0.0478|0.5168|-0.1569|0.3047|
> ||iTransformer|0.0161|0.1031|0.0383|0.2278|0.0102|0.0985|-0.1496|0.0682|
> ||TRA|0.0260|0.1813|0.0464|0.3285|0.0504|0.6040|-0.1461|0.3450|
> |**Factor Libraries**|Alpha 158|0.0192|0.1353|0.0374|0.2639|0.0199|0.2515|-0.1771|0.1124|
> ||Alpha 360|0.0195|0.1331|0.0308|0.2089|0.0191|0.2527|-0.1270|0.1504|
> ||AutoAlpha|0.0184|0.1529|0.0175|0.1382|0.0397|0.5728|-0.1006|0.3946|
> |**RD-Agent Series**|RD-Factor$_{\text{GPT-4o}}$|0.0201|0.1709|0.0176|0.1404|0.1010|1.3730|-0.0787|1.2833|
> ||RD-Factor$_{\text{o4-mini}}$|0.0264|**0.2652**|0.0345|0.3454|0.0849|1.0014|-0.1215|0.6985|
> ||RD-Model$_{\text{GPT-4o}}$|0.0259|0.1649|0.0532|0.3469|0.1039|1.0941|-0.1367|0.7600|
> ||RD-Model$_{\text{o4-mini}}$|$\underline{0.0265}$|0.1825|0.0521|**0.3616**|   0.1160|1.4021|$\underline{-0.0735}$|1.5777|
> ||RD-Agent(Q)$_{\text{GPT-4o}}$|0.0241|0.1532|$\underline{0.0555}$|$\underline{0.3574}$|$\underline{0.1358}$|$\underline{1.4227}$|-0.0803|$\underline{1.6903}$|
> ||RD‑Agent(Q)$_{\text{o4-mini}}$|**0.0288**|$\underline{0.1828}$|**0.0564**|0.3523|**0.1982**|**2.1721**|**-0.0656**|**3.0229**|
>
> As shown in the above table, these results further support that $\text{RD-Agent(Q)}$ maintains strong performance even under **forward-looking**, **out-of-training-distribution** market conditions.
>
> ---
>
> **For *W4* (Stochasticity of $G$):**
>
> > The mathematical function G defined in line 129 is not strictly deterministic due to the nonzero temperature setting of the LLM, which implies stochastic outputs rather than a deterministic mapping.
>
> We thank the reviewer for pointing out that $G$ is not strictly deterministic due to the LLM's nonzero temperature. Our intention was to present $G$ as a conceptual abstraction of the research process, but we agree that, strictly speaking, $G$ is stochastic. We will clarify this in the revision.
>
> ---
>
> **For *F1* (Figure Labeling):**
>
> > Figure 1 lacks clear stage labels, which affects readability.
>
> We thank the reviewer for the helpful suggestion regarding Figure 1. We will add labels to the pipeline in the revision to improve clarity.
>
> ---
>
> **For *F2* (CoT Citation):**
>
> > Chain-of-thought should be cited in line 75.
>
> We thank the reviewer for pointing this out. We will add the citation in the revision.
>
> ---
>
> **For *F3* (LLM-driven Units):**
>
> > Section 2 does not explicitly indicate which units are LLM-driven or how LLMs are applied, making it unclear to readers whether these are implemented via supervised learning, reinforcement learning, or LLMs without reading the Appendix.
>
> We thank the reviewer for pointing this out and apologize for the confusion. We agree that the components in Section 2 lack clear indication that they are LLM-driven. We will make the following changes:
>
> - At the beginning of Section 2, we will explicitly change the sentence "*into five tightly coupled units*" to "*into five tightly LLM-powered coupled units that mainly gather information and react with LLM API calls at their core*."
> - In Figure 3, we will add a robot icon before each component name to align with Section 1 and draw further attention to this point.
> - In the remainder of Section 2, we will also add this information in each component's paragraph.
>
> We hope these changes will mitigate the confusion for the reviewer and for future readers.

---

### Official Review · Reviewer_heC6 · 2025-07-06

**Rating:** 3
**Confidence:** 3

**Summary:**

The paper proposed an iterative, multi-agent framework that seeks to automate the process of quantitative research and development. Experiments conducted in the paper show the proposed method can achieve higher performance in backtesting.

**Additional Feedback:**

The paper used historic data (no newer than 2020) for backtesting to evaluate the performance of the proposed method. These data are likely seen by the LLM during training. Is look-ahead bias a concern here? If so, how to deal with it?

**Dataset Code Accessibility:**

Yes

**Ethical Considerations:**

No, there are no or only very minor ethics concerns

**Limitations Weaknesses:**

novelty: the paper described a framework with five key components that seek to provide iterative and automated qualitative research. While the proposed method is interesting, it is unclear what might be some of the key novel contributions made in the paper that leads to the performance of the proposed method. It is also not clear how these novel contributions contrast against existing work.

relevance: I am not sure how to justify the relevance of this paper to the Datasets and Benchmarks set. The paper appear to proposed  a new algorithm/framework for trading in financial markets. It seems to be more appropriate for the main track.

**Strengths Contributions:**

significance: quantitative  research is time consuming and expensive. the paper proposed a multi-agent framework to automate the quantitative research process. As a result, the paper tackles a significant yet challenging problem.

potential impact: it makes sense that quantitative research can be more automated in the future, and language model based agent can play a role in this process.

---

> ### Author Rebuttal · Authors · 2025-07-31
>
> We sincerely thank the reviewer for the thorough and detailed feedback. Below are our responses.
>
> **For *Significance* & *Potential Impact*:**
>
> We appreciate the reviewer's support for our framework. We would like to further emphasize that, to the best of our knowledge, $\text{RD-Agent(Q)}$ is **the first** end-to-end framework in quantitative finance to fully automate factor-model code research and development **based entirely on data**, with iterative automation. This approach helps address key challenges of LLMs, such as *hallucination* and *poor interoperability*, which have contributed to its broad adoption in both academia and industry.
>
> ---
>
> **For *Novelty Weakness*:**
>
> > It is unclear which key novel components drive the improved performance of our framework, and how these contributions distinguish themselves from existing work.
>
> To conclude, it is very hard to identify the **most** novel component because all components are irreplaceable in the system. We agree with Reviewer `Sif6` that our key contribution is system integration rather than a specific algorithmic advancement. However, we do not treat the workflow as a black box, so we have conducted extensive investigations into each component's superiority.
>
> Every component in our system has its own job and uniqueness:
> - Specification Unit: defines the high-level task context;
> - Synthesis Unit: generates new hypotheses and concrete implementation tasks based on past experiments and domain knowledge;
> - Implementation Unit: converts tasks into executable code;
> - Validation Unit: conducts experiments in realistic scenarios;
> - Analysis Unit: evaluates results and guides iterative improvements.
>
> By clearly separating the R&D stages, we achieve **modular and interpretable automation**. We investigate each component's superiority through extensive experiments against existing methods:
> - Figure 5 demonstrates the superior performance of the Synthesis Unit during the research phase, illustrating the evolutionary refinement of the final SOTA factor library under our framework.
> - Figures 6, 7, and 8, and Table 3 confirm the excellence of the Implementation Unit's Co-STEER agent compared to other code generation methods.
> - Table 2 presents ablation studies on various action selection strategies, validating the advantage of our contextual two-armed bandit scheduler.
>
> Moreover, at the framework comparison level, as noted earlier, $\text{RD-Agent(Q)}$ is the first fully data-centric framework in quantitative finance that automates factor-model code R&D with iterative refinement. Due to this unique scope, direct end-to-end comparisons against existing methods are challenging. Nonetheless, we provide factor- and model-level comparisons against strong baselines, with multiple experiments (Table 1, Figures 7 and 8) demonstrating consistent superiority and robustness.
>
> ---
>
> **For *Relevance to DB track*:**
>
> > It is unclear how the submission fits within the scope of the Datasets and Benchmarks (D&B) track, as it appears to propose a new algorithm/framework for trading that might be more suited for the main track.
>
> Actually, our work aligns perfectly with the NeurIPS 2025 Datasets and Benchmarks (D\&B) track, especially with the recently expanded scope emphasizing:
>
> **“Data-centric AI methods and tools, e.g., to measure and improve data quality or utility, or studies in data-centric AI that bring important new insight.”**
>
> Quantitative finance is a domain where data quality, factor construction, and model robustness are vital. $\text{RD-Agent(Q)}$ explicitly addresses these data-centric challenges by automating factor and model research, thereby improving the **efficiency and effectiveness of data utilization** for quantitative strategy development.
>
> Our framework's cost-efficiency is demonstrated by the low experimental cost (under \$10 per full experiment cycle; see Appendix D.4), highlighting its practicality for data-centric R\&D.
>
> Moreover, the framework is **generalizable** beyond finance: by adapting dataset constraints and domain priors, it can be applied to a wide range of data science tasks, including time series forecasting, computer vision, and natural language processing benchmarks (e.g., Kaggle competitions).
>
> ---
>
> **For *Additional Feedback*:**
>
> > The paper used historic data (no newer than 2020) for backtesting to evaluate the performance of the proposed method. These data are likely seen by the LLM during training. Is look-ahead bias a concern here? If so, how to deal with it?
>
> This is an important question. Our framework leverages a data-centric approach that provides distinct advantages. Specifically, we do not expose the LLM to real-time data or explicit dataset splits; rather, the LLM only receives the data schema and volume information, as shown in the prompt for the Specification Unit (Appendix E.1). This design prevents the LLM from knowing precise temporal boundaries or training-validation-test partitions, thereby effectively mitigating information leakage concerns.
>
> To further address this issue, we conducted additional experiments on the **CSI 500 dataset** with a clearly separated and more recent time split:
> * Train: 2008-01-01 to 2021-12-31
> * Validation: 2022-01-01 to 2023-12-31
> * Test: 2024-01-01 to 2025-06-30
>
> For LLM backends, we use
> - `GPT-4o` (training cutoff: 2023-10-01, which is **completely** prior to our test period)
> - `o4-mini` (training cutoff: 2024-06-01, which is **almost entirely** prior to our test period)
>
> *(Due to temporary API limitations, `o3-mini` was replaced by `o4-mini`, which has comparable capacity and performance.)*
>
> Below are the experimental results demonstrating the robustness of our method under this strict temporal partition:
>
> ### **Experiments on CSI 500 and test from 2024 (Ori. Table 1)**
> |Model Type| Model|IC|ICIR|Rank IC|Rank ICIR|ARR|IR (SHR\*)|MDD|CR|
> | :- | :- | :-: | :-: | :-: | :-: | :-: | :-: | :-:| :-: |
> |**Machine-Learning**|LightGBM|0.0181|0.1271|0.0393|0.2783|-0.0294|-0.3178|-0.2089|-0.1407|
> ||XGBoost|0.0240|0.1675|0.0427|0.3054|0.0053|0.0634|-0.1766|0.0300|
> ||CatBoost|0.0241|0.1629|0.0390|0.2627|0.0111|0.1438|-0.1799|0.0617|
> ||DoubleEnsemble|0.0248|0.1705|0.0423|0.2850|0.0227|0.2500|-0.2094|0.1084|
> |**Deep-Learning**|Transformer|0.0194|0.1355|0.0416|0.2884|0.0234|0.2898|-0.1331|0.1758|
> ||GRU|0.0188|0.1022|0.0512|0.2711|0.0398|0.3716|-0.1602|0.2484|
> ||LSTM|0.0219|0.1434|0.0401|0.2825|0.0560|0.6900|-0.1075|0.5209|
> ||GATs|0.0162|0.1013|0.0426|0.2731|0.0478|0.5168|-0.1569|0.3047|
> ||iTransformer|0.0161|0.1031|0.0383|0.2278|0.0102|0.0985|-0.1496|0.0682|
> ||TRA|0.0260|0.1813|0.0464|0.3285|0.0504|0.6040|-0.1461|0.3450|
> |**Factor Libraries**|Alpha 158|0.0192|0.1353|0.0374|0.2639|0.0199|0.2515|-0.1771|0.1124|
> ||Alpha 360|0.0195|0.1331|0.0308|0.2089|0.0191|0.2527|-0.1270|0.1504|
> ||AutoAlpha|0.0184|0.1529|0.0175|0.1382|0.0397|0.5728|-0.1006|0.3946|
> |**RD-Agent Series**|RD-Factor$_{\text{GPT-4o}}$|0.0201|0.1709|0.0176|0.1404|0.1010|1.3730|-0.0787|1.2833|
> ||RD-Factor$_{\text{o4-mini}}$|0.0264|**0.2652**|0.0345|0.3454|0.0849|1.0014|-0.1215|0.6985|
> ||RD-Model$_{\text{GPT-4o}}$|0.0259|0.1649|0.0532|0.3469|0.1039|1.0941|-0.1367|0.7600|
> ||RD-Model$_{\text{o4-mini}}$|$\underline{0.0265}$|0.1825|0.0521|**0.3616**|   0.1160|1.4021|$\underline{-0.0735}$|1.5777|
> ||RD-Agent(Q)$_{\text{GPT-4o}}$|0.0241|0.1532|$\underline{0.0555}$|$\underline{0.3574}$|$\underline{0.1358}$|$\underline{1.4227}$|-0.0803|$\underline{1.6903}$|
> ||RD‑Agent(Q)$_{\text{o4-mini}}$|**0.0288**|$\underline{0.1828}$|**0.0564**|0.3523|**0.1982**|**2.1721**|**-0.0656**|**3.0229**|
>
> These results confirm that $\text{RD-Agent(Q)}$ performs robustly on **truly out-of-distribution**, **unseen data**, providing strong evidence against look-ahead bias and information leakage.

---

### Note · Authors · 2025-08-12

We sincerely thank the reviewers and the NeurIPS 2025 committee for your time and thoughtful feedback. We are pleased to see that our work, RD-Agent(Q), has received strong recognition from the reviewer and the community. We received about 7K GitHub stars after we opensource our project which means he proposed framework has generated significant interest in the open-source ecosystem, and our experimental results are both solid and reproducible.

We appreciated all reviewers about their recognition. Reviewer *rHYN* noted that our approach is “**novel and insightful**” and highlighted its potential for “**far-reaching impact beyond quantitative finance.**” Reviewer *Sif6* praised the “**modular, extensible architecture**” and “**solid empirical benchmarking,**” while Reviewer *heC6* acknowledged that our work “**tackles a significant yet challenging problem**” and “**makes sense for future automation in quantitative research.**”

The primary concern raised by reviewers was the limited scope of our original experiments, which focused only on the CSI 300 dataset and a specific time period. In response, we conducted extensive additional experiments on both the CSI 500 and NASDAQ 100 datasets, using strictly out-of-distribution time splits. These results demonstrate the robustness and generalizability of our framework. Reviewer rHYN explicitly stated that “**concerns on evaluation methods have been adequately addressed,**” and increased their final score to 5. Reviewer Sif6 also acknowledged the improvements but provided no feedback. Since all reviewers shared very similar concern, **we believe our extensive experiments should resolve the main concerns of ALL reviewers**.

However, we regret that the rebuttal phase did not lead to meaningful engagement from all reviewers. Notably, Reviewer heC6 did not acknowledge or respond to our rebuttal and appeared unaware of the updated DB track scope, and there was minimal participation in the discussion overall. We believe this lack of interaction limited the opportunity to clarify misunderstandings and strengthen the review process.

Finally, we would like to express our sincere gratitude to the Area Chairs, Senior Area Chairs, and Program Chairs for overseeing the review process. We hope our responses and additional experiments will be taken into consideration during the final decision.

---

### Decision · Program_Chairs · 2025-09-18

**Decision:**

Accept (poster)

**Comment:**

This submission presents RD‑Agent(Q), a data‑centric multi‑agent framework that automates the full R&D loop for quantitative strategies via five LLM‑driven stages, a structured code‑generation agent (Co‑STEER), and a contextual bandit scheduler for factor–model co‑optimization; the paper reports consistent gains over classical factor libraries and deep time‑series baselines, and adds out‑of‑sample studies during rebuttal. Reviewers broadly agree the problem is important and the architecture is modular and extensible. After rebuttal—where the authors added CSI‑500 and NASDAQ‑100 experiments with test periods in 2024–2025 and clarified leakage/recency concerns. On balance, the main contribution is system integration rather than new learning algorithms, but the work is substantive, reproducible (released code/data), and squarely data‑centric, fitting the D&B track. For camera‑ready, the authors should (i) emphasize integrative novelty and clearly mark LLM‑powered units while clarifying the stochastic nature of G, (ii) broaden related work/baselines and keep the added cross‑market results, and (iii) fix minor dataset‑metadata issues flagged by the automated report (missing core metadata and one inaccessible file).